# Mitigating the Participation Bias by Balancing Extreme Ratings

## Abstract

Rating aggregation plays a crucial role in various fields, such as product recommendations, hotel rankings, and teaching evaluations. However, traditional averaging methods can be affected by participation bias, where some raters do not participate in the rating process, leading to potential distortions. In this paper, we consider a robust rating aggregation task under the participation bias. We assume that raters may not reveal their ratings with a certain probability depending on their individual ratings, resulting in partially observed samples. Our goal is to minimize the expected squared loss between the aggregated ratings and the average of all underlying ratings (possibly unobserved) in the worst-case scenario.

We focus on two settings based on whether the sample size (i.e. the number of raters) is known. In the first setting, where the sample size is known, we propose an aggregator, named as the Balanced Extremes Aggregator. It estimates unrevealed ratings with a balanced combination of extreme ratings. When the sample size is unknown, we derive another aggregator, the Polarizing-Averaging Aggregator, which becomes optimal as the sample size grows to infinity. Numerical results demonstrate the superiority of our proposed aggregators to participation bias, compared to simple averaging.

## CCS Concepts

• **Theory of computation** → **Algorithmic game theory**; **Algorithmic mechanism design**.

## Keywords

Rating Aggregation, Participation Bias, Robust Aggregation

**ACM Reference Format:**
Anonymous Author(s). 2018. Mitigating the Participation Bias by Balancing Extreme Ratings. In *Proceedings of Make sure to enter the correct conference title from your rights confirmation email (Conference acronym 'XX)*. ACM, New York, NY, USA, 14 pages. https://doi.org/XXXXXXX.XXXXXXX

## 1 Introduction

Suppose you come back from a hotel stay that left you frustrated. The service was terrible, and the noise kept you up all night. The experience was bad enough that, as soon as you got home, you felt compelled to go online and leave a negative review. This is not something you normally do—in fact, the last time you stayed at a hotel that was just okay, you did not even bother to rate it.

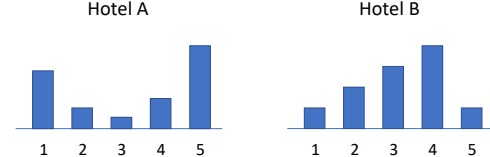

**Figure 1: The observed ratings of two hotels.**

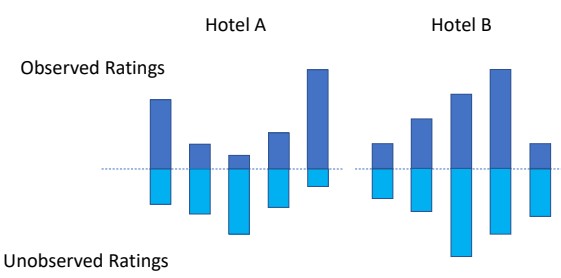

**Figure 2: The true ratings of two hotels.**

Now, imagine you're on the other side, trying to decide between two hotels on an online platform. Both have identical average ratings, but as shown in Figure 1, their rating distributions are quite different. Which one would you pick?

While many people rely on average ratings to make their choices, these scores can be misleading due to biases in the way they are collected. One such bias is participation bias, where the ratings do not represent the views of all guests—e.g., only the ones who felt strongly enough to leave feedback [5, 57].

At Hotel A, reviews are polarized: some guests, possibly those staying in the premium rooms, report excellent stays with 5-star ratings, while others leave very negative feedback. In contrast, Hotel B's reviews are consistently around 4 stars, indicating generally positive but not outstanding experiences. This suggests that Hotel A's displayed rating might be inflated, as guests with neutral opinions may have skipped reviewing, whereas Hotel B's balanced ratings offer a more accurate reflection. Figure 2 shows a possible way of the true distribution.

Such participation bias affects other areas too, like course evaluations, movie reviews, and product ratings on e-commerce platforms, where only extreme opinions tend to be represented [5, 25, 27, 57].

If we could estimate the likelihood of participation across different rater groups, we could apply traditional methods like importance sampling to approximate the true average rater ratings [45, 46, 49]. However, without data from non-participants, quantifying the precise extent of this bias remains challenging. Therefore, we need to use alternative metrics to evaluate the aggregation methods without full knowledge of the participation bias. Following the

field of robustness analysis [3], we employ a robust aggregation paradigm aimed at minimizing the worst-case error to mitigate the participation bias.

## 1.1 Problem Statement

For all positive integer $k$, $[k]$ denotes the set $\{1, 2, \cdots, k\}$.

We aim to find a robust function $f$ to aggregate the discrete ratings from $n$ raters. Each rating $x_i \in [m]$ follows an underlying distribution $\boldsymbol{p} = (p_1, p_2, \cdots, p_m)$ and we want to estimate the expectation $\mu = \mathbb{E}_{x \sim \boldsymbol{p}}[x]$. We assume the ratings are independent. Otherwise, the actual ratings may contain systematic biases, making it impossible for any aggregator to recover the true expectation even given all ratings. Under this assumption, the best estimator for $\mu$ is the empirical average of all ratings $\frac{1}{n} \sum_i x_i$.

However, due to the participation bias, raters will report their ratings with probabilities that depend on the rating values. Let $\boldsymbol{g} = (g_1, g_2, \ldots, g_m)$ represent these probabilities, where $g_r$ is the probability of an rater reporting her rating when it is $r$. We assume raters are homogeneous and have the same participation bias. As a result, only a subset of elements in $\mathbf{X}$ is observed, which we denote by $\hat{\mathbf{X}}$. For each rater $i$:

$$\hat{x}_i = \begin{cases} x_i & \text{with probability } g_{x_i} \\ 0 & \text{otherwise} \end{cases}$$

where 0 indicates that the rating is unobserved. Let $D_{\boldsymbol{p}}$ denote the distribution generating $\mathbf{X}$ and $D_{\boldsymbol{p},\boldsymbol{g}}$ denote the distribution generating $\hat{\mathbf{X}}$. We also call $D_{\boldsymbol{p},\boldsymbol{g}}$ the information structure.

Without any information on the likelihood of reporting, the worst-case scenario would be no one reports, resulting in a lack of any useful aggregator. Thus, we assume there exists a known parameter $q$, indicating the lowest reporting probability among raters. That is, $g_r \in [q, 1]$ for any rating $r$. When $q = 1$, there is no participation bias and as $q$ decreases, the bias may increase accordingly. We use an example to illustrate our model.

EXAMPLE 1.1 (WHY DIRECT AVERAGE IS NOT GOOD). *Consider a binary rating system where $x_i \in \{1, 2\}$ with equal probability. Here, let $g_1 = 1$ and $g_2 = q$, meaning that negative reviews ($x_i = 1$) are always observed, while positive reviews ($x_i = 2$) are observed only with probability $q$.*

*Under this setup, even with a large sample size $n$, the direct average of observed ratings will tend toward $\frac{1+2q}{1+q}$, which significantly underestimates the true expectation of $3/2$ when $q$ is small.*

We measure the performance of an aggregator by its expected squared error, defined as $L(f, \boldsymbol{p}, \boldsymbol{g}) = \mathbb{E}_{\hat{\mathbf{X}} \sim D_{\boldsymbol{p},\boldsymbol{g}}}\left[(f(\hat{\mathbf{X}}) - \mu)^2\right]$. One objective could be finding $f$ that minimizes the worst-case error: $\max_{\boldsymbol{p},\boldsymbol{g}} L(f, \boldsymbol{p}, \boldsymbol{g})$. However, when the sample size $n$ is small, selecting $\boldsymbol{p}$ with high variances can lead to large errors even for the ideal aggregator who observes $\boldsymbol{x}$. Since we focus on addressing the errors attributable to uncertainties in $\boldsymbol{g}$, we adopt a robust approach and instead minimize the regret relative to the ideal aggregator who observes the full data $\boldsymbol{x}$, and outputs $\frac{1}{n} \sum_i x_i$:

$$\min_f \max_{\boldsymbol{p},\boldsymbol{g}} \mathbb{E}_{\mathbf{X} \sim D_{\boldsymbol{p}}, \hat{\mathbf{X}} \sim D_{\boldsymbol{p},\boldsymbol{g}}}\left[(f(\hat{\mathbf{X}}) - \mu)^2 - \left(\frac{1}{n} \sum_i x_i - \mu\right)^2\right].$$

For short, we denote $R(f, \boldsymbol{p}, \boldsymbol{g}) = \mathbb{E}_{\mathbf{X} \sim D_{\boldsymbol{p}}, \hat{\mathbf{X}} \sim D_{\boldsymbol{p},\boldsymbol{g}}}\left[(f(\hat{\mathbf{X}}) - \mu)^2 - (\frac{1}{n} \sum_i x_i - \mu)^2\right]$.

Because we have assumed the raters are homogeneous, observing $\hat{\boldsymbol{x}}$ is equivalent as observing the histograms $n_u, n_1, n_2, \cdots, n_m$ where $n_r$ counts the number of ratings of $r$ for all $r \in \{1, \cdots, m\}$, and $n_u$ indicates the number of unobserved ratings. That is, the aggregator's input is $n_u, n_1, n_2, \cdots, n_m$. This is applicable in the scenarios when the sample size $n$ is known. For example, in the teaching evaluation, the number of students is known to the instructors but not all students will give feedback.

*A variant with unknown $n$.* However, in practice, we may not know $n$. For example, in the movie rating platforms, we cannot obtain the number of people watching the movies. Thus, we consider a variant of the above problem where $n$ is unknown. In this case, the aggregation problem remains the same except that the aggregator's input is $n_1, n_2, \cdots, n_m$.

## 1.2 Summary of Results

We explore the optimal aggregator in two settings, depending on the knowledge of the sample size $n$ (i.e., the number of raters).

*The sample size $n$ is known.* In Section 3, we assume $n$ is known. We construct a lower bound for the regret by considering a mixture of two specific information structures. Based on the lower bound, we provide a new aggregator, the Balanced Extreme Aggregator (BEA), which is the best response to those two specific information structures. Intuitively, BEA estimates the expected ratings of those unreported raters based on the difference between $n_1$ and $n_m$, the counts of extreme ratings. Then BEA adjusts the observed rating average with the estimated unobserved rating average. Figure 3 illustrates the whole process. We illustrate its near-optimal performance numerically in various cases.

*The sample size $n$ is unknown.* In Section 4, we assume $n$ is unknown. Since we do not know how many unobserved ratings there are, the above aggregator is not applicable. Instead, we obtain a new aggregator, the Polarizing-Averaging Aggregator (PAA), and show its optimality when $n$ goes to infinite. Furthermore, we numerically show that PAA performs well for a finite sample size. Figure 4 illustrates the whole process. We create two modified histograms from the original observed data. For the first histogram, we identify a threshold $k_1$, and only keep $q$ fraction of the counts for ratings above $k_1$. In the second histogram, we identify a threshold $k_2$, and only keep $q$ fraction of the counts for ratings below $k_2$. We then calculate the empirical mean for each of these adjusted histograms and output their average.

Section 5 shows the virtualization of our aggregators, which helps to establish some insights about the aggregators.

## 2 Related Literature

*Rating Aggregation.* Rating aggregation can be defined as the problem of integrating different ratings into a single representative value. Previous works aim to resist individual disturbance from random or malicious spammers [4, 6] by building a reputation system [14, 26, 40]. The basic idea of reputation system is weighted average where a rater's weight is measured by his reputation score

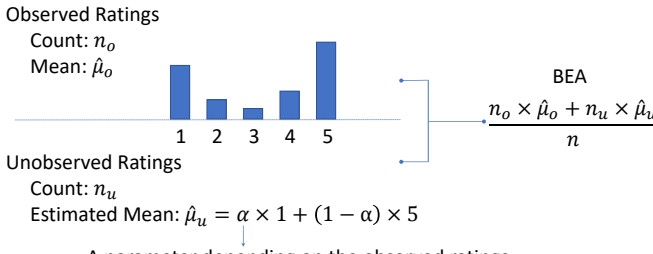

**Figure 3: Illustration of the Balanced Extreme Aggregator (BEA) when the sample size $n$ is known. BEA estimates the expected ratings of those unreported raters based on the difference between $n_1$ and $n_m$, the counts of extreme ratings. Then BEA adjusts the observed rating average with the estimated unobserved rating average.**

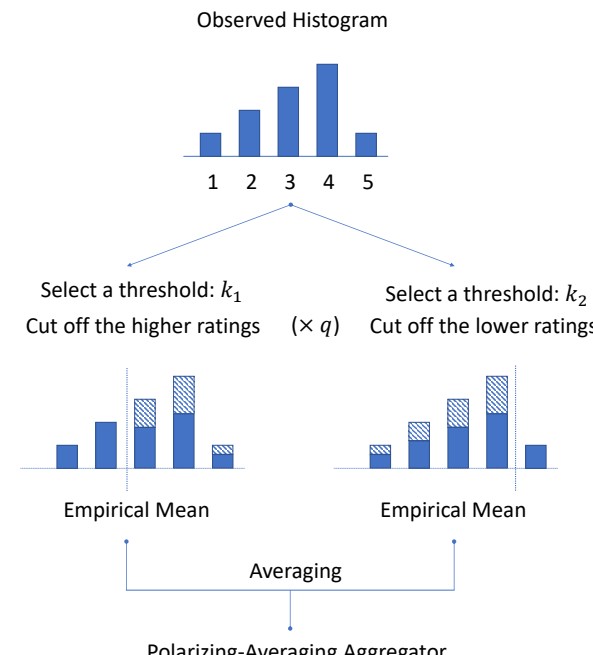

**Figure 4: Illustration of the Polarizing-Averaging Aggregator (PAA) when the sample size $n$ is unknown. We create two modified histograms from the original observed data. For the first histogram, we identify a threshold $k_1$, and only keep $q$ fraction of the counts for ratings above $k_1$. In the second histogram, we identify a threshold $k_2$, and only keep $q$ fraction of the counts for ratings below $k_2$. We then calculate the empirical mean for each of these adjusted histograms and output their average.**

which is calculated by using the rater-object weighed bipartite network. The quality-based methods measure a raters's reputation by

the difference between the rating values and the objects' weighted average rating values [29, 33, 54, 56]. Unlike quality-based methods, raters are grouped based on their rating similarities and their reputation is calculated by the corresponding group sizes in group-based methods. [13, 15, 16] Another branch of literature considers how to resist collusive disturbance where malicious raters simultaneously promote or demote the qualities of the targeted objects by giving consistent rating score [36, 50–52, 55].

Our work focuses on the simple setting where there is a single object so we can not use the rater-object weighed bipartite network to learn raters' rating behavioral patterns. We only consider participation bias rather than other malicious behaviors. Finally, we adopt an adversarial approach to evaluate the performance of an aggregation scheme.

*Participation Bias.* Participation bias is a well-studied topic within the casual inference community which arises when participants disproportionately possess certain traits that affect participation [11]. It is a common source of error in clinical trials and survey studies [7, 8, 12, 42, 43].

When participation bias is only related to some observed (measured) variables and other representative samples are available (e.g. the whole population data from government), weighting strategies make it possible to create a pseudo-sample representative for these variables [41, 42] under the assumption that non-participants have equivalent behaviours to participants in the same socio-demographic category. An alternative way is multiple imputation (MI), which is viable when data are missing at random [2, 39, 44] and can be strengthened if we can infer information on the absent non-participants using record-linkage [17, 19] or other methods. If the data are thought to be missing-not-at-random then methods involving sensitivity analyses like pattern mixture modelling [34] and NARFCS [47] can be helpful. These methods can also be combined to correct participation bias [18].

Methods used to correct sample selection bias can also be used to correct participation bias [48].

The first solution to sample selection bias was suggested by Heckman [21] who proposed a maximum likelihood estimator which heavily relies on the normality assumption. One way to relax normality while remaining in the maximum likelihood framework was suggested by Lee [30, 31]. Two-Step estimators are frequently employed in empirical work including parametric models [22, 23] and semi-parametric models [1, 24, 32].

Recent works on sample selection models have aimed to address robust alternatives to the Heckman model. Marchenko and Genton [35] introduced a selection-t model, which models the error distribution using a Student's t distribution for dealing with the robustness to the normal assumption in the Heckman model. Zhelonkin et al. [53] proposed a modified robust semi-parametric alternative based on Heckman's two-step estimation method with asymptotic normality and gave its asymptotic variance. de Souza Bastos et al. [10] propose a generalization of the Heckman sample selection model by allowing the sample selection bias and dispersion parameters to depend on covariates.

Our work significantly differs from the above literature in a few aspects. First, we are only interested in recovering the average score rather than the casual effect. Second, we consider a simplified

setting where we only know the reported score and the lower bound of participation rate instead of various socio-demographic data. Finally, instead of making extra assumptions, we adopt a robust approach to evaluate the performance of an aggregation scheme.

*Robust Aggregation.* Arieli et al. [3] first introduced a robust paradigm for forecast aggregation, aiming to minimize the aggregator's regret compared to an omniscient aggregator. Later, De Oliveira et al. [9] explored a robust absolute paradigm focused on minimizing the aggregator's own loss. Since then, a growing number of researches have addressed robust information aggregation under various information structures, including the projective substitutes condition [37], conditional independent setting [3], and scenarios involving second-order information [38]. Furthermore, Guo et al. [20] provide an algorithmic approach for computing a near-optimal aggregator across general information structures. Kong et al. [28] tackle robust aggregation in the context of base rate neglect, a specific bias in forecast aggregation.

We consider a different setting from the above literature. We focus on addressing participation bias in rating aggregation.

## 3 Rating Aggregation with Known Sample Size

This section will discuss the setting with a known number of agents $n$. We provide the Balanced Extremes Aggregator (BEA), which performs near-optimal numerically in various cases.

BEA estimates the expectation of the unobserved ratings, $\hat{\mu}_u$, using a convex combination of extreme ratings, given by: $\hat{\mu}_u = \alpha \times 1 + (1-\alpha) \times m$ where $\alpha$ is a parameter that depends on the observed ratings. Intuitively, $\alpha$ increases with the number of observed ratings of 1. BEA then combines $\hat{\mu}_u$ with the average observed ratings to produce a final aggregated result.

DEFINITION 3.1 (BALANCED EXTREMES AGGREGATOR). *Given sample size $n$, the observed ratings $\hat{x}$, the aggregator's output is defined as $f^{BEA}(\hat{x}) = \frac{n_o \hat{\mu}_o + n_u \hat{\mu}_u}{n}$. Here $n_o = \sum_i \mathbb{1}(\hat{x}_i \neq 0)$ is the number of observed ratings, $n_u = n - n_o$ is the number of unobserved ratings. $\hat{\mu}_o = \frac{\sum_i \hat{x}_i}{n_o}$ is the average of the observed ratings, $\hat{\mu}_u = \alpha \times 1 + (1-\alpha) \times m$ is the estimation of the unobserved ratings' expectation. $\alpha$ [1] depends on the difference between the count of ratings of 1 and the count of ratings of $m$, denoted as $n_1 - n_m$, and the lower bound of the participation probability, denoted as $q$.*

The design of BEA is based on a specific mixture of information structures which helps to establish a lower bound for the regret. In particular, solving $\min_f \max_{p,g} R(f, p, g)$ is equivalent to solving a zero-sum game between nature who selects an information structure $(p, g)$, and a decision-maker who selects the aggregator $f$. We will present a specific mixed strategy of the nature, and use this mixed strategy to establish a lower bound for the regret. BEA is the best response to this mixed strategy.

LEMMA 3.2 (LOWER BOUND OF THE REGRET). *Consider a pair of information structures:*

---

[1] $\alpha = \frac{(a^*q)^{n_1-n_m}}{(a^*q)^{n_1-n_m}+(1-a^*)^{n_1-n_m}}$. $a^*$ is the solution to

$$\max_a \sum_{s,t} \binom{n}{t}\binom{n-t}{s} a^{n-t}(1-a)^t q^s (1-q)^{n-s-t} \left( \frac{(n-s-t)(m-1)(1-a)^{s-t}}{n((aq)^{s-t}+(1-a)^{s-t})} \right)^2$$

.

- $\theta_1 = (p_1 = [a, 0, \cdots, 0, 1-a], g_1 = [q, 1, \cdots, 1, 1])$
- $\theta_2 = (p_2 = [1-a, 0, \cdots, 0, a], g_2 = [1, 1, \cdots, 1, q])$

*where $q$ is the lower bound of the participation probability, and $a \in [0, 1]$ is a parameter. For any aggregator $f$, the regret has a lower bound*

$$\max_{p,g} R(f, p, g)$$
$$\geq \max(R(f, p_1, g_1), R(f, p_2, g_2))$$
$$\geq \sum_{s,t} \binom{n}{t}\binom{n-t}{s} a^{n-t}(1-a)^t q^s (1-q)^{n-s-t} \left( \frac{(n-s-t)(m-1)(1-a)^{s-t}}{n((aq)^{s-t}+(1-a)^{s-t})} \right)^2.$$

*Proof Sketch.* We provide a proof sketch here, and the complete proof is deferred to Appendix A.

Consider a mixture of these two information structures: $\theta = \theta_1$ or $\theta_2$ with equal probability. Then given the observed ratings $\hat{X}$, we can compute the posterior $\Pr[\theta = \theta_1 | \hat{X}]$. The best predictor for the expectation of unobserved ratings will be the conditional expectation $\mathbb{E}[x|\hat{x} = 0]$ given the information structure is $\theta$. Then we can calculate the regret of the best predictor to obtain our lower bound.

By enumerating all possible $a$, we obtain a corollary

COROLLARY 3.3. *For any aggregator $f$,*

$$\max_{p,g} R(f, p, g)$$
$$\geq \max_a \sum_{s,t} \binom{n}{t}\binom{n-t}{s} a^{n-t}(1-a)^t q^s (1-q)^{n-s-t} \left( \frac{(n-s-t)(m-1)(1-a)^{s-t}}{n((aq)^{s-t}+(1-a)^{s-t})} \right)^2$$

Let $a^* = \arg\max_a \sum_{s,t} \binom{n}{t}\binom{n-t}{s} a^{n-t}(1-a)^t q^s (1-q)^{n-s-t} \left( \frac{(n-s-t)(m-1)(1-a)^{s-t}}{n((aq)^{s-t}+(1-a)^{s-t})} \right)^2$, then according to our proof of Lemma 3.2, BEA is the best response to $\theta_1(a^*), \theta_2(a^*)$. That is, for BEA,

$$R(f, \theta_1(a^*)) = R(f, \theta_2(a^*))$$
$$= \max_a \sum_{s,t} \binom{n}{t}\binom{n-t}{s} a^{n-t}(1-a)^t q^s (1-q)^{n-s-t} \left( \frac{(n-s-t)(m-1)(1-\mu)}{n} \right)^2$$

Based on Corollary 3.3, BEA outperforms any other aggregator in our lower bound situation. Regarding general situations, we evaluate the performance of BEA numerically using Matlab. Figure 5 shows the results, where the grey line is the theoretical lower bound given by Corollary 3.3, the red line is BEA and the blue line is the simple averaging. BEA is almost optimal for a wide range of the parameter $q$, and outperforms simple averaging especially when $q$ is small. When $q$ is close to 1, indicating there is almost no bias, BEA is not optimal. We will discuss more details in Appendix B.

## 4 Rating Aggregation with Unknown Sample Size

This section will discuss the case where the number of agents $n$ is unknown. Fortunately, when $n \to \infty$, meaning that we have a large number of agents, we can obtain a closed-form of the optimal aggregator, which is called the Polarizing-Averaging Aggregator.

When $n$ is unknown, the aggregator only has access to the observed histogram $n_1, n_2, \cdots, n_m$. PAA outputs the average of the empirical mean of two modified histograms. For the first histogram, we identify a threshold $k_1$, and only keep $q$ fraction of the counts for ratings above $k_1$. In the second histogram, we identify a threshold $k_2$, and only keep $q$ fraction of the counts for ratings below $k_2$.

Notice that in the described process, only the empirical mean of histograms is relevant. Therefore, we can use the normalized

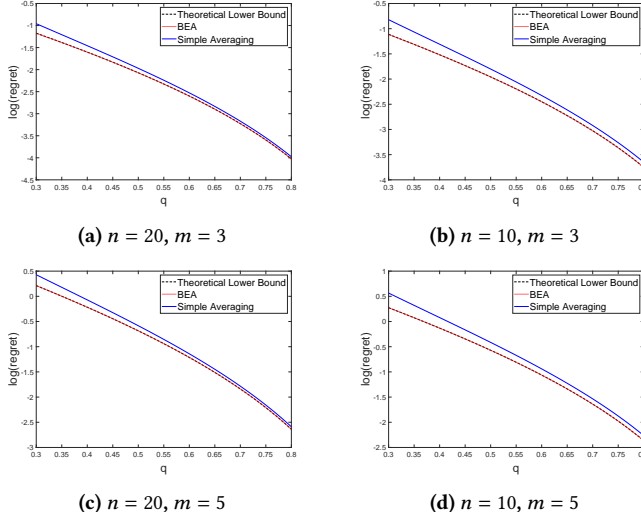

**(a)** $n = 20$, $m = 3$      **(b)** $n = 10$, $m = 3$

**(c)** $n = 20$, $m = 5$      **(d)** $n = 10$, $m = 5$

Figure 5: Simple averaging vs. BEA for different sample size $n$ and the number of rating categories $m$. The x-axis is the lower bound of the participation probability, $q$, and the y-axis is the natural logarithm of the regret. The regret of BEA almost matches the theoretical lower bound for a wide range of $q$.

histograms, specifically the empirical distribution, $\hat{p}_r = \frac{n_r}{\sum_j n_j}, \forall r \in [m]$, as input.

DEFINITION 4.1 (POLARIZING-AVERAGING AGGREGATOR). *Define the Polarizing-Averaging Aggregator* $f^{PAA}(\hat{p}) = \frac{u(\hat{p})+l(\hat{p})}{2}$, *where*

$$l(\hat{p}) = \frac{\sum_{r=1}^{k_1(\hat{p})} r \times \hat{p}_r + q \sum_{r=k_1(\hat{p})+1}^{m} r \times \hat{p}_r}{\sum_{r=1}^{k_1(\hat{p})} \hat{p}_r + q \sum_{r=k_1(\hat{p})+1}^{m} \hat{p}_r}2,$$

$$u(\hat{p}) = \frac{q \sum_{r=1}^{k_2(\hat{p})} r \times \hat{p}_r + \sum_{r=k_2(\hat{p})+1}^{m} r \times \hat{p}_r}{q \sum_{r=1}^{k_2(\hat{p})} \hat{p}_r + \sum_{r=k_2(\hat{p})+1}^{m} \hat{p}_r}3.$$

We will show that $l(\hat{p})$ is the lower bound of the true expectation $\mu$ given the empirical distribution $\hat{p}$ and the lower bound of the participation probability $q$, and $u(\hat{p})$ is the upper bound.

## 4.1 Analysis of PAA in the asymptotic case

This section will analyze PAA in the asymptotic case. We first show that in the asymptotic case, PAA is optimal. We will then present the regret and the worst information structure of PAA in the asymptotic case.

---
2

$$k_1(\hat{p}) = \arg \max_{1 \le k \le m} \sum_{r=1}^{k-1} (r - k)\hat{p}_r + q \sum_{r=k+1}^{m} (r - k)\hat{p}_r \ge 0$$

3

$$k_2(\hat{p}) = \arg \max_{1 \le k \le m} q \sum_{r=1}^{k-1} (r - k)\hat{p}_r + \sum_{r=k+1}^{m} (r - k)\hat{p}_r \ge 0.$$

### 4.1.1 Optimality of PAA.

THEOREM 4.2. *When* $n \to \infty$, *PAA is optimal.*

*Proof Sketch.* We provide a proof sketch here, the complete proof is deferred to Appendix A.

We first notice that when $n \to \infty$, the regret is the loss. When $n \to \infty$, the ideal aggregator who observes full ratings knows the true distribution $p$. Therefore, the regret of $f$ to the ideal aggregator equals the loss $\max_{p,g}(f(\hat{p}) - \mathbb{E}_{x\sim p}[x])^2$. The problem becomes solving $\min_f \max_{p,g}(f(\hat{p}) - \mathbb{E}_{x\sim p}[x])^2$ conditional on knowing the empirical distribution of the observed ratings $\hat{p}$.

Furthermore, when $n \to \infty$, $\hat{p}$ is propositional to the element-wise product[4] of the true distribution $p$ and the participation probabilities $g$: $\hat{p} \propto p \circ g$.

The proof is divided into the following steps.

*Optimal aggregator is the midpoint of extremes.* Given $\hat{p}$, define the lower bound of the expectation of $p$ which satisfy the propositional constraint $l^*(\hat{p}) = \min_{p,g:\hat{p}\propto p\circ g} \mathbb{E}_{x\sim p}[x]$ and the upper bound $u^*(\hat{p}) = \max_{p,g:\hat{p}\propto p\circ g} \mathbb{E}_{x\sim p}[x]$.

Since $(f(\hat{p})-\mathbb{E}_{x\sim p}[x])^2$ measures the squared distance to $\mathbb{E}_{x\sim p}[x]$, the best aggregator must be the midpoint between the extreme values to minimize this squared distance. That is, the best $f$ has the format of $(u^*(\hat{p}) + l^*(\hat{p}))/2$.

It's left to show that $l^*(\hat{p}) = l(\hat{p})$ and $u^*(\hat{p}) = u(\hat{p})$ where $l(\hat{p})$ and $u(\hat{p})$ are described in the definition of PAA (Definition 4.1).

*Minimize or maximize* $\mathbb{E}_{x\sim p}[x]$ *given* $\hat{p}$. Given $\hat{p}$, we start to find $(p', g')$ to minimize or maximize $\mathbb{E}_{x\sim p}[x]$, conditional on $\hat{p} \propto p \circ g$. We first characterize $g'$. Here are two properties of the optimal $g'$.

- $g'$ is extreme: $g_r = 1$ or $q$ for any rating $r$.
- $g'$ is monotonic: $g_r \le g_{r+1}$ or $g_r \ge g_{r+1}$ for any rating $r$.

Here is how we derive the above properties. Given $\hat{p}$, because $\hat{p} \propto p \circ g$, $\mathbb{E}_{x\sim p}[x]$ can be viewed as a function, denoted as $F$, of $g$. By calculating the partial derivative of $F$, we observe the sign of $\frac{dF}{dg_r}$ is independent with $g_r$ as long as $g_r$ is positive. The sign of $\frac{dF}{dg_r}$ can be determined by $F(g) - r$.

Using these two properties of $g'$, $g'$ can be constructed as follows: for the minimum, there exists an index $k_1(\hat{p})$ such that $g_r = q$ for any $r \le k_1(\hat{p})$ and $g_r = 1$ otherwise. While for the maximum, there exists an index $k_2(\hat{p})$ such that $g_r = 1$ for any $r \le k_2(\hat{p})$ and $g_r = q$ otherwise.

To find the optimal $k_1(\hat{p}), k_2(\hat{p})$, we can simply enumerate all the $m$ values, but we make a more careful analysis to give the closed-form of the optimal $k_1(\hat{p}), k_2(\hat{p})$ in the appendix.

This finishes the proof sketch of the optimality of PAA in the asymptotic case.

### 4.1.2 Regret and the worst information structure of PAA.
We also obtain the closed forms of regret and the corresponding worst information structure of PAA.

PROPOSITION 4.3. *When* $n \to \infty$, *the regret of PAA is* $\left(\frac{(m-1)(1-q)}{2(1+q)}\right)^2$. *In addition, the corresponding worst pair of information structure is*

- $\theta_1 = (p_1 = [\frac{1}{q+1}, 0, \cdots, 0, \frac{q}{q+1}], g_1 = [q, 1, \cdots, 1, 1])$

---
[4]For any vectors $x, y$, their element-wise product is $x \circ y = (x_1 y_1, \cdots, x_n y_n)$.

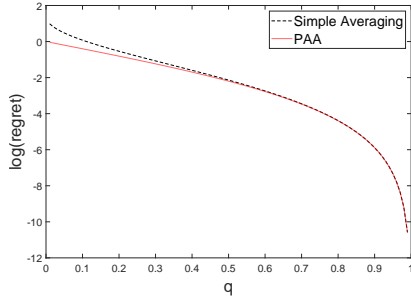

**Figure 6: Simple averaging vs. PAA for a specific case where the number of rating categories is $m = 3$: The x-axis is the lower bound of the participation probability, $q$, and the y-axis is the natural logarithm of the regret.**

- $\theta_2 = (\boldsymbol{p}_2 = [\frac{q}{q+1}, 0, \cdots, 0, \frac{1}{q+1}], \boldsymbol{g}_2 = [1, 1, \cdots, 1, q])$

*That is,*

$$R(f^{PAA}, \theta_1) = R(f^{PAA}, \theta_2) = \left( \frac{(m-1)(1-q)}{2(1+q)} \right)^2 .$$

*Proof Sketch.* Fix the optimal aggregator PAA, we aim to find the worst information structure $(\boldsymbol{p}^*, \boldsymbol{g}^*)$ that maximizes the loss/regret $(f^{PAA}(\hat{\boldsymbol{p}}) - \mathbb{E}_{x\sim\boldsymbol{p}}[x])^2$.

We have proved that the optimal aggregator is the midpoint of the extremes $l(\hat{\boldsymbol{p}})$ and $u(\hat{\boldsymbol{p}})$. The maximal loss is $(u(\hat{\boldsymbol{p}}) - l(\hat{\boldsymbol{p}}))^2 / 4$. To obtain the maximal loss, we aim to find $\hat{\boldsymbol{p}}^*$ to maximize $u(\hat{\boldsymbol{p}}) - l(\hat{\boldsymbol{p}})$. We analyze the maximizer in the following four steps.

- We first show that $\hat{\boldsymbol{p}}^*$ has at most three non-zero entries: $|\{r : \hat{p}_r > 0\}| \leq 3$.
- We further show that $\hat{\boldsymbol{p}}^*$ has exactly two non-zero entries: $|\{r : \hat{p}_r^* > 0\}| = 2$.
- We then obtain the concrete value: $\hat{\boldsymbol{p}}^* = (\frac{1}{2}, 0, \cdots, 0, \frac{1}{2})$.
- Finally, we construct two worst information structures

$$\theta_1 = (\boldsymbol{p}_1 = [\frac{1}{q+1}, 0, \cdots, 0, \frac{q}{q+1}], \boldsymbol{g}_1 = [q, 1, \cdots, 1, 1]),$$

$$\theta_2 = (\boldsymbol{p}_2 = [\frac{q}{q+1}, 0, \cdots, 0, \frac{1}{q+1}], \boldsymbol{g}_2 = [1, 1, \cdots, 1, q]).$$

In particular, $\theta_1$ is the maximizer of $\mathbb{E}_{x\sim\boldsymbol{p}}[x]$ given $\hat{\boldsymbol{p}} = \hat{\boldsymbol{p}}^*$. $\theta_2$ is the minimizer of $\mathbb{E}_{x\sim\boldsymbol{p}}[x]$ given $\hat{\boldsymbol{p}} = \hat{\boldsymbol{p}}^*$.

**First Step:** For any empirical distribution $\hat{\boldsymbol{p}}$, recall that the participation probabilities in the maximizer of $\mathbb{E}_{x\sim\boldsymbol{p}}[x]$ given $\hat{\boldsymbol{p}}$ has the format that $g_r = 1$ for any $r \leq k_1(\hat{\boldsymbol{p}})$ and $g_r = q$ otherwise. The participation probabilities in the minimizer of $\mathbb{E}_{x\sim\boldsymbol{p}}[x]$ given $\hat{\boldsymbol{p}}$ has the format that $g_r = q$ for any $r \leq k_2(\hat{\boldsymbol{p}})$ and $g_r = 1$ otherwise.

The aim is to maximize the gap $u(\hat{\boldsymbol{p}}) - l(\hat{\boldsymbol{p}})$. We first divide $\hat{\boldsymbol{p}}$ into three components by $k_1(\hat{\boldsymbol{p}}) \leq k_2(\hat{\boldsymbol{p}})$. Then we prove that by alternately concentrating all probabilities in the first component at rating 1 and all probabilities in the third component at rating $m$, we can achieve a new distribution with a non-decreasing gap. Note that in the new distribution, the threshold will also change. We repeat this concentration step until the distribution stabilizes at $\hat{\boldsymbol{p}}'$, which has the following form: $(\hat{p}'_1, 0, \cdots, 0, \hat{p}'_{k_1(\hat{\boldsymbol{p}}')+1}, \cdots, \hat{p}'_{k_2(\hat{\boldsymbol{p}}')}, 0, \cdots, 0, \hat{p}'_m)$. At last we concentrate all probabilities in the second component of $\hat{\boldsymbol{p}}'$ at one of rating $(k_1(\hat{\boldsymbol{p}}') + 1)$ and rating $k_2(\hat{\boldsymbol{p}}')$ to obtain a new

distribution with a non-decreasing gap and at most three non-zero entries.

**Second Step:** Now we have proved that $\hat{\boldsymbol{p}}^*$ has at most three non-zero entries at position $1, k \in (1, m), m$. We further find that we can redistribute the density at the middle position $k$ to one of the end positions $1, m$ will not decrease the gap $u(\hat{\boldsymbol{p}}) - l(\hat{\boldsymbol{p}})$. Thus, we show that $\hat{\boldsymbol{p}}^*$ has exactly two non-zero entries: $|\{r : \hat{p}_r^* > 0\}| = 2$.

**Third Step:** Given the simple format of $\hat{\boldsymbol{p}}^*$, we obtain its value by differentiation.

**Fourth Step:** Finally, we calculate the worst information structures $(\boldsymbol{p}^*, \boldsymbol{g}^*)$ given $\hat{\boldsymbol{p}}^*$ by computing the maximizer and the minimizer of $\mathbb{E}_{x\sim\boldsymbol{p}}[x]$ given $\hat{\boldsymbol{p}} = \hat{\boldsymbol{p}}^*$.

## 4.2 Analysis of PAA in the finite case

PAA is also applicable for the finite case. Though PAA is not optimal when $n$ is finite, we prove that it is near-optimal, with the error bound depending on the sample size $n$, and the number of rating categories $m$.

**THEOREM 4.4.** *When* $m = o\left( \left( \frac{n}{\ln n} \right)^{\frac{1}{4}} \right)$, *PAA is* $O\left( m^2 \sqrt{\frac{\ln n}{n}} \right)$-*optimal. That is, for any aggregator $f$,*

$$\max_{\boldsymbol{p}, \boldsymbol{g}} R(f^{PAA}, \boldsymbol{p}, \boldsymbol{g}) \leq \max_{\boldsymbol{p}, \boldsymbol{g}} R(f, \boldsymbol{p}, \boldsymbol{g}) + O\left( m^2 \sqrt{\frac{\ln n}{n}} \right).$$

*Proof Sketch.* The input to $f^{PAA}$ is the empirical distribution of observed ratings. In the asymptotic scenario, this corresponds to $\boldsymbol{p} \circ \boldsymbol{g}$, and we have demonstrated that $f^{PAA}$ is optimal under these conditions. However, in finite sample situations, the empirical distribution is only an approximate version of $\boldsymbol{p} \circ \boldsymbol{g}$. Consequently, the performance of $f^{PAA}$ hinges on the quality of this approximation. We will further analyze the performance by employing the Chernoff bound and union bound to assess the approximation's accuracy.

Furthermore, when $n$ is finite, unlike the asymptotic case, the regret is not the loss. When $n$ is different, the ideal aggregator is also different. The analysis should also consider this.

## 5 Virtualization of Aggregators

*Visualization of BEA.* Figure 7 and Figure 8 visualize BEA for a specific case where the sample size $n = 20$, and the rating values are $[m] = [2] = \{1, 2\}$. Compared to simple averaging, BEA tends to be more conservative (i.e. close to $\frac{m+1}{2} = 3/2$). Note as the lower bound of the participation probability $q$ becomes larger, BEA becomes more conservative.

*Visualization of PAA.* Figure 9 and Figure 10 visualize PAA for a specific case where the sample size $n = 1000$, and the rating values are $[m] = [2] = \{1, 2\}$. Note as the lower bound of the participation probability $q$ becomes smaller, PAA becomes more conservative (i.e. close to $\frac{m+1}{2} = 3/2$). We can see it more clearly in Figure 10.

When $n_1 = n_2 = 300$, all the aggregators report the same value $\frac{m+1}{2} = \frac{3}{2}$. In general, $f^{BEA}$ also reports $\frac{m+1}{2}$ when $n_1 = n_m$. When $n$ is large, the worst case is roughly $n_1 = n_m$, when $n \rightarrow \infty$, the worst $\hat{\boldsymbol{p}}$ is $(\frac{1}{2}, 0, \cdots, 0, \frac{1}{2})$. This represents a polarized situation where participants are split between extreme approval or disapproval, with equal numbers on both sides.

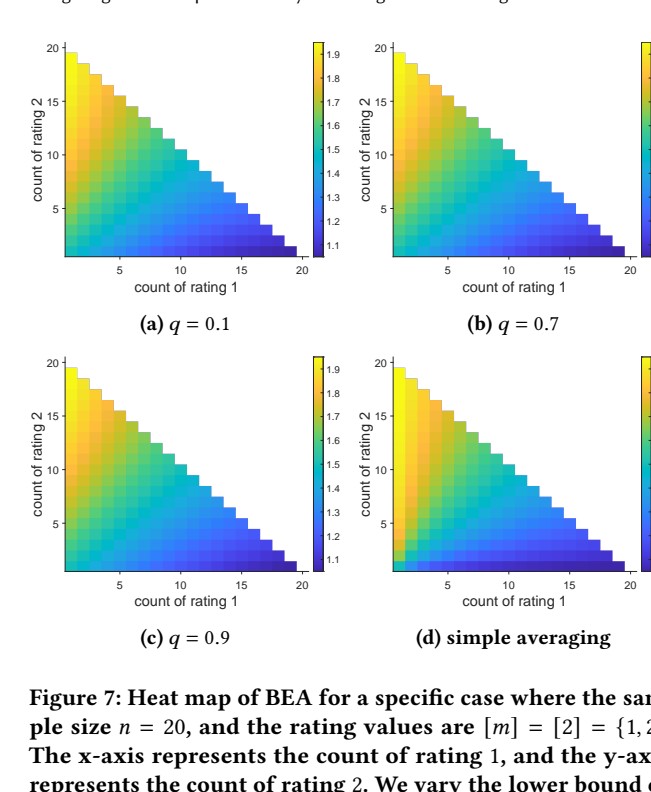

**(a)** $q = 0.1$      **(b)** $q = 0.7$

**(c)** $q = 0.9$      **(d)** simple averaging

**Figure 7: Heat map of BEA for a specific case where the sample size $n = 20$, and the rating values are $[m] = [2] = \{1, 2\}$. The x-axis represents the count of rating $1$, and the y-axis represents the count of rating $2$. We vary the lower bound of the participation probability $q$.**

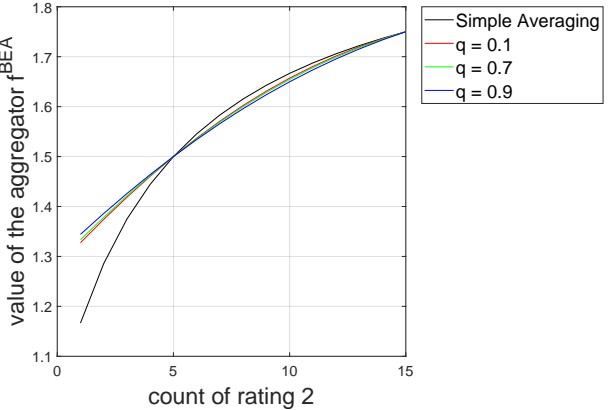

**Figure 8: Value of BEA for a specific case where the sample size $n = 20$, and the rating values are $[m] = [2] = \{1, 2\}$. The count of rating $1$ is fixed as $n_1 = 5$. The x-axis is the count of rating $2$, $n_2$. The y-axis is the value of BEA. We vary the lower bound of the participation probability $q$.**

## 6 Conclusion

In this work, we explored the problem of rating aggregation with the participation. We focus on two scenarios where the sample size may be either known or unknown. For the known case, we introduced the Balanced Extremes Aggregator (BEA), which balances the extreme ratings by predicting the unobserved ratings

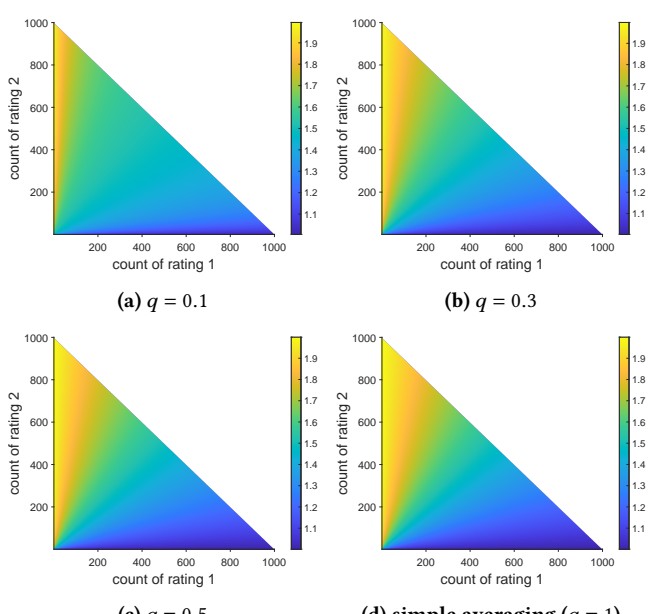

**(a)** $q = 0.1$      **(b)** $q = 0.3$

**(c)** $q = 0.5$      **(d)** simple averaging ($q = 1$)

**Figure 9: Heat map of PAA for a specific case where the sample size $n = 1000$, and the rating values are $[m] = [2] = \{1, 2\}$. The x-axis represents the count of rating $1$, and the y-axis represents the count of rating $2$. We vary the lower bound of the participation probability $q$. When $q = 1$, PAA degenerates to simple averaging.**

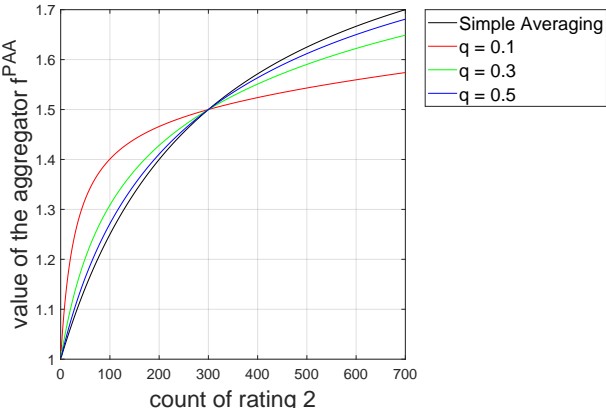

**Figure 10: Value of PAA for a specific case where the sample size $n = 1000$, and the rating values are $[m] = [2] = \{1, 2\}$. The count of rating $1$ is fixed as $n_1 = 300$. The x-axis is the count of rating $2$, $n_2$. The y-axis is the value of PAA. We vary the lower bound of the participation probability $q$. When $q = 1$, PAA degenerates to simple averaging.**

with the observed ratings. We evaluate its performance by numerical results. For the unknown sample size setting, we presented the Polarizing-Averaging Aggregator (PAA), which achieves near-optimal performance by averaging two polarized true distributions given the observed ratings.

As for the future work, we could explore adaptive algorithms that dynamically adjust to varying participation probabilities over time could enhance robustness, especially in rapidly changing environments like e-commerce and social media platforms.

In this work, we assume the ratings are independent. Incorporating user behavior analysis, such as identifying and adjusting for systematic biases based on user demographics or past behavior, could yield more general aggregators.

Another promising direction involves extending our aggregators to handle multi-dimensional ratings, which are common in modern applications where users rate multiple attributes (e.g., service, quality, and price).

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

## A  Omitted Proofs

**Lemma A.1 (Lower Bound of the Regret).** *Consider a pair of information structures:*

- $\theta_1 = (\boldsymbol{p}_1 = [a, 0, \cdots, 0, 1-a], \boldsymbol{g}_1 = [q, 1, \cdots, 1, 1])$
- $\theta_2 = (\boldsymbol{p}_2 = [1-a, 0, \cdots, 0, a], \boldsymbol{g}_2 = [1, 1, \cdots, 1, q])$

*where $a$ is a parameter. Then for any aggregator $f$, the regret has a lower bound*

$$\max_{\boldsymbol{p},\boldsymbol{g}} R(f, \boldsymbol{p}, \boldsymbol{g})$$
$$\geq \max(R(f, \boldsymbol{p}_1, \boldsymbol{g}_1), R(f, \boldsymbol{p}_2, \boldsymbol{g}_2))$$
$$\geq \sum_{s,t} \binom{n}{t}\binom{n-t}{s} a^{n-t}(1-a)^t q^s (1-q)^{n-s-t} \left( \frac{(n-s-t)(m-1)(1-a)^{s-t}}{n((aq)^{s-t}+(1-a)^{s-t})} \right)^2 .$$

**Proof.** Consider a linear combination information structure $\theta = \theta_1$ or $\theta_2$ with equal probability. Now we compute its best response $f'(\hat{\mathbf{X}}) = \mathbb{E}\left[\frac{1}{n}\sum_i x_i | \hat{\mathbf{X}}\right]$, which is the posterior given the observed ratings. Given $s = \sum_i \mathbb{1}(\hat{x}_i = 1), t = \sum_i \mathbb{1}(\hat{x}_i = m)$. We have

$$\Pr[\theta = \theta_1 | \hat{\mathbf{X}}]$$
$$= \frac{\Pr[\hat{\mathbf{X}}|\theta_1]}{\Pr[\hat{\mathbf{X}}|\theta_1] + \Pr[\hat{\mathbf{X}}|\theta_2]}$$
$$= \frac{\binom{n}{t}\binom{n-t}{s} a^{n-t}(1-a)^t q^s (1-q)^{n-s-t}}{\binom{n}{t}\binom{n-t}{s}a^{n-t}(1-a)^t q^s (1-q)^{n-s-t} + \binom{n}{s}\binom{n-s}{t} a^{n-s}(1-a)^s q^t (1-q)^{n-s-t}}$$
$$= \frac{(aq)^{s-t}}{(aq)^{s-t}+(1-a)^{s-t}} .$$
$$= \frac{1}{1+(\frac{1-a}{aq})^{s-t}}$$

Then

$$\mathbb{E}[x|\hat{x}=0]$$
$$= \Pr[\theta = \theta_1 | \hat{\mathbf{X}}]\mathbb{E}[x|\hat{x}=0, \theta_1] + \Pr[\theta = \theta_2|\hat{\mathbf{X}}]\mathbb{E}[x|\hat{x}=0, \theta_2]$$
$$= 1 * \frac{1}{1+(\frac{1-a}{aq})^{s-t}} + m * (1 - \frac{1}{1+(\frac{1-a}{aq})^{s-t}})$$
$$= m - (m-1)\frac{1}{1+(\frac{1-a}{aq})^{s-t}} .$$

Thus the best response is $f'(\hat{\mathbf{X}}) = \frac{\sum_i \hat{x}_i + (m-(m-1)\frac{1}{1+(\frac{1-a}{aq})^{s-t}}) * (n-s-t)}{n}$.
By simple calculation we obtain that for any $f$,
$$R(f, \Theta) \geq R(f, \theta) \geq R(f', \theta)$$
$$= \sum_{s,t}\binom{n}{t}\binom{n-t}{s}a^{n-t}(1-a)^t q^s (1-q)^{n-s-t}\left(\frac{(n-s-t)(m-1)(1-\mu)}{n}\right)^2 \quad \square$$

**Theorem A.2.** *When $n \to \infty$, the aggregator is optimal.*

**Proof.** When $n \to \infty$, the empirical distribution is propositional to the element-wise product of the true distribution $\boldsymbol{p}$ and the participation probabilities $\boldsymbol{g}$: $\hat{\boldsymbol{p}} \propto \boldsymbol{p} \circ \boldsymbol{g}$. We want to minimize $\max_{\boldsymbol{p},\boldsymbol{g}}(f(\hat{\boldsymbol{p}}) - \mathbb{E}_{x\sim\boldsymbol{p}}[x])^2$.

Given $\hat{\boldsymbol{p}}$, define the lower bound of the expectation of $\boldsymbol{p}$ which satisfy the propositional constraint $l^*(\hat{\boldsymbol{p}}) = \min_{\boldsymbol{p},\boldsymbol{g}:\hat{\boldsymbol{p}}\propto\boldsymbol{p}\circ\boldsymbol{g}} \mathbb{E}_{x\sim\boldsymbol{p}}[x]$ and the upper bound $u^*(\hat{\boldsymbol{p}}) = \max_{\boldsymbol{p},\boldsymbol{g}:\hat{\boldsymbol{p}}\propto\boldsymbol{p}\circ\boldsymbol{g}} \mathbb{E}_{x\sim\boldsymbol{p}}[x]$. Since $(f(\hat{\boldsymbol{p}}) - \mathbb{E}_{x\sim\boldsymbol{p}}[x])^2$ is a quadratic function about $\mathbb{E}_{x\sim\boldsymbol{p}}[x]$, we have

$$\max_{\boldsymbol{p},\boldsymbol{g}}(f(\hat{\boldsymbol{p}}) - \mathbb{E}_{x\sim\boldsymbol{p}}[x])^2 = \max\{(f(\hat{\boldsymbol{p}}) - l^*(\hat{\boldsymbol{p}}))^2, (f(\hat{\boldsymbol{p}}) - u^*(\hat{\boldsymbol{p}}))^2\}$$

$$\geq (u^*(\hat{\boldsymbol{p}}) - l^*(\hat{\boldsymbol{p}}))^2/4.$$

The equality holds if and only if $f(\hat{\boldsymbol{p}}) = (u^*(\hat{\boldsymbol{p}})+l^*(\hat{\boldsymbol{p}}))/2$, so the best aggregator is $f(\hat{\boldsymbol{p}}) = (u^*(\hat{\boldsymbol{p}}) + l^*(\hat{\boldsymbol{p}}))/2$. It's left to show that $l^*(\hat{\boldsymbol{p}}) = l(\hat{\boldsymbol{p}})$ and $u^*(\hat{\boldsymbol{p}}) = u(\hat{\boldsymbol{p}})$ where $l(\hat{\boldsymbol{p}})$ and $u(\hat{\boldsymbol{p}})$ are described in the definition of PAA (Definition 4.1).

Given $\hat{\boldsymbol{p}}$, we start to find $(\boldsymbol{p}', \boldsymbol{g}')$ to minimize or maximize $\mathbb{E}_{x\sim\boldsymbol{p}}[x]$, conditional on $\hat{\boldsymbol{p}} \propto \boldsymbol{p} \circ \boldsymbol{g}$. We first characterize $\boldsymbol{g}'$. Here are two properties of the optimal $\boldsymbol{g}'$.

- $\boldsymbol{g}'$ is extreme: $g_r = 1$ or $q$ for any rating $r$.
- $\boldsymbol{g}'$ is monotonic: $g_r \leq g_{r+1}$ or $g_r \geq g_{r+1}$ for any rating $r$.

Given $\hat{\boldsymbol{p}}$, because $\hat{\boldsymbol{p}} \propto \boldsymbol{p} \circ \boldsymbol{g}$, $\mathbb{E}_{x\sim\boldsymbol{p}}[x]$ can be viewed as a function, denoted as $F$, of $\boldsymbol{g}$. By calculating the partial derivative of $F$, we have

$$\frac{dF}{dg_r} = \frac{p_r \hat{g}_r^{-2}}{(\sum_j \hat{p}_j/g_j)^2} * (\sum_j j\frac{\hat{p}_j}{g_j} - r\sum_j \frac{\hat{p}_j}{g_j})$$
$$= \frac{p_r \hat{g}_r^{-2}}{(\sum_j \hat{p}_j/g_j)^2} * (\sum_{j=1}^{r-1}(j-r)\frac{\hat{p}_j}{g_j} + \sum_{j=r+1}^m (j-r)\frac{\hat{p}_j}{g_j})$$
$$= \frac{p_r \hat{g}_r^{-2}}{\sum_j \hat{p}_j/g_j} * (F(\boldsymbol{g}) - r).$$

First Notice that the sign of $\frac{dF}{dg_r}$ is independent with $g_r$ as long as $g_r$ is positive, so the optimal $\boldsymbol{g}$ is extreme: $g_r = 1$ or $g_r = q$. Then Notice the sign of $\frac{dF}{dg_r}$ is determined by $F(\boldsymbol{g}) - r$ where $F(\boldsymbol{g}) = \mathbb{E}_{x\sim\boldsymbol{p}}[x] \in [1, m]$, so the optimal $\boldsymbol{g}$ is monotonic.

Using these two properties of $\boldsymbol{g}'$, $\boldsymbol{g}'$ can be given by an index strategy:

for the minimum, there exists an index $k_1(\hat{\boldsymbol{p}})$ such that

$$g'_r = \begin{cases} q & r \leq k_1(\hat{\boldsymbol{p}}) \\ 1 & r > k_1(\hat{\boldsymbol{p}}) \end{cases}$$

while for the maximum, there exists an index $k_2(\hat{\boldsymbol{p}})$ such that

$$g'_r = \begin{cases} 1 & r \leq k_2(\hat{\boldsymbol{p}}) \\ q & r > k_2(\hat{\boldsymbol{p}}) \end{cases}$$

To find the optimal index, we can simply calculate all the $m$ values, but here we make a more careful analysis.

Take the minimum for example. We use $\boldsymbol{g}^{(k)}$ to denote a $1 \times m$ vector with the first $k$ elements being $q$ and the other elements being 1. We claim $F(\boldsymbol{g}^{(k)})$ has a single-bottom shape regarding the index $k$, or equivalently,

$$F(\boldsymbol{g}^{(k)}) - F(\boldsymbol{g}^{(k-1)}) > 0 \implies \forall k' \geq k,\ F(\boldsymbol{g}^{(k')}) - F(\boldsymbol{g}^{(k'-1)}) \geq 0.$$

Let $D_k = \frac{1}{q}\sum_{i=1}^{k}\hat{p}_i + \sum_{i=k+1}^{m}\hat{p}_i$. We have

$$F(\boldsymbol{g}^{(k)}) - F(\boldsymbol{g}^{(k-1)}) = (\frac{1}{q} - 1)\hat{p}_k D_k(k - F(\boldsymbol{g}^k)).$$

We first prove $F(\boldsymbol{g}^{(k)}) < k \implies \forall k' \geq k,\ F(\boldsymbol{g}^{(k')}) < k'$. Notice

$$F(\boldsymbol{g}^{(k+1)}) - F(\boldsymbol{g}^{(k)}) = (\frac{1}{q} - 1)\hat{p}_{k+1} D_{k+1}((k+1) - F(\boldsymbol{g}^{k+1})).$$

So

$$F(\boldsymbol{g}^{(k+1)}) = \frac{F(k) + (\frac{1}{q} - 1)(k+1)\hat{p}_{k+1}D_{k+1}}{1 + (\frac{1}{q} - 1)\hat{p}_{k+1}D_{k+1}}.$$

If $F(\boldsymbol{g}^{(k)}) < k$, we have

$$\begin{aligned} F(g^{k+1}) &= \frac{F(k) + (\frac{1}{q} - 1)(k+1)\hat{p}_{k+1}D_{k+1}}{1 + (\frac{1}{q} - 1)\hat{p}_{k+1}D_{k+1}} \\ &< \frac{k + (\frac{1}{q} - 1)(k+1)\hat{p}_{k+1}D_{k+1}}{1 + (\frac{1}{q} - 1)\hat{p}_{k+1}D_{k+1}} \\ &< \frac{(k+1) + (\frac{1}{q} - 1)(k+1)\hat{p}_{k+1}D_{k+1}}{1 + (\frac{1}{q} - 1)\hat{p}_{k+1}D_{k+1}} \\ &= k+1 \end{aligned}$$

By induction, we have $\forall k' \geq k,\ F(\boldsymbol{g}^{(k')}) < k'$. So

$$\begin{aligned} F(\boldsymbol{g}^{(k)}) - F(\boldsymbol{g}^{(k-1)}) > 0 &\implies F(\boldsymbol{g}^{(k)}) < k \\ &\implies \forall k' \geq k,\ F(\boldsymbol{g}^{(k')}) < k' \\ &\implies \forall k' \geq k,\ F(\boldsymbol{g}^{(k')}) - F(\boldsymbol{g}^{(k'-1)}) \geq 0. \end{aligned}$$

So $k_1(\hat{\boldsymbol{p}})$ is the optimal index for minimum $\iff F(\boldsymbol{g}^{(k_1(\hat{\boldsymbol{p}}))}) - k_1(\hat{\boldsymbol{p}}) > 0, F(\boldsymbol{g}^{(k_1(\hat{\boldsymbol{p}})+1)}) - (k_1(\hat{\boldsymbol{p}}) + 1) < 0$ ( Since $\hat{\boldsymbol{p}}$ can have zero entries, there might be some consecutive $k$ that all of them is "optimal". For clarity, we take this definition). i.e.

$$k_1(\hat{\boldsymbol{p}}) = \max\{k : 1 \leq k \leq m, \sum_{i=1}^{k-1}(i-k)\frac{\hat{p}_i}{q} + \sum_{i=k+1}^{m}(i-k)\hat{p}_i \geq 0\}.$$

Similarly,

$$k_2(\hat{\boldsymbol{p}}) = \max\{k : 1 \leq k \leq m, \sum_{i=1}^{k-1}(i-k)\hat{p}_i + \sum_{i=k+1}^{m}(i-k)\frac{\hat{p}_i}{q} \geq 0\}.$$

Notice the optimal index is also a rough approximation of the optimal value. Since $k_1(\hat{\boldsymbol{p}}) < l(\hat{\boldsymbol{p}}) < k_1(\hat{\boldsymbol{p}}) + 1, k_2(\hat{\boldsymbol{p}}) < u(\hat{\boldsymbol{p}}) < k_2(\hat{\boldsymbol{p}}) + 1$ and $l(\hat{\boldsymbol{p}}) < u(\hat{\boldsymbol{p}})$, we have $k_1(\hat{\boldsymbol{p}}) \leq k_2(\hat{\boldsymbol{p}})$. □

PROPOSITION A.3. *When $n \to \infty$, the regret of PAA is $\left(\frac{(m-1)(1-q)}{2(1+q)}\right)^2$. In addition, the corresponding worst pair of information structure is*

- $\theta_1 = (\boldsymbol{p}_1 = [\frac{1}{q+1}, 0, \cdots, 0, \frac{q}{q+1}], \boldsymbol{g}_1 = [q, 1, \cdots, 1, 1])$
- $\theta_2 = (\boldsymbol{p}_2 = [\frac{q}{q+1}, 0, \cdots, 0, \frac{1}{q+1}], \boldsymbol{g}_2 = [1, 1, \cdots, 1, q])$

*That is,*

$$R(f^{PAA}, \theta_1) = R(f^{PAA}, \theta_2) = \left(\frac{(m-1)(1-q)}{2(1+q)}\right)^2.$$

PROOF. Fix the optimal aggregator PAA, we aim to find the worst information structure $(\boldsymbol{p}^*, \boldsymbol{g}^*)$ that maximizes the loss/regret $(f^{PAA}(\hat{\boldsymbol{p}}) - \mathbb{E}_{x \sim \boldsymbol{p}}[x])^2$.

We have proved that the optimal aggregator is the midpoint of the extremes $l(\hat{\boldsymbol{p}})$ and $u(\hat{\boldsymbol{p}})$. The maximal loss is $(u(\hat{\boldsymbol{p}}) - l(\hat{\boldsymbol{p}}))^2/4$. To obtain the maximal loss, we aim to find $\hat{\boldsymbol{p}}^*$ to maximize $u(\hat{\boldsymbol{p}}) - l(\hat{\boldsymbol{p}})$. We analyze the maximizer in the following four steps.

First we prove $\hat{\boldsymbol{p}}^*$ has at most three non-zero entries. Then we prove $\hat{\boldsymbol{p}}^*$ has exactly two non-zero entries. Next we prove the $\hat{\boldsymbol{p}}^*$ is $(\frac{1}{2}, 0, \cdots, 0, \frac{1}{2})$. Finally, we calculate the worst information structures $(\boldsymbol{p}^*, \boldsymbol{g}^*)$ given $\hat{\boldsymbol{p}}^*$ by computing the maximizer and the minimizer of $\mathbb{E}_{x \sim \boldsymbol{p}}[x]$ given $\hat{\boldsymbol{p}} = \hat{\boldsymbol{p}}^*$.

**First Step:** For any $\hat{\boldsymbol{p}}$, consider the optimal index for minimum $k_1(\hat{\boldsymbol{p}})$ and the optimal index for maximum $k_2(\hat{\boldsymbol{p}})$. Since $k_1(\hat{\boldsymbol{p}}) < l(\hat{\boldsymbol{p}}) < k_1(\hat{\boldsymbol{p}}) + 1, k_2(\hat{\boldsymbol{p}}) < u(\hat{\boldsymbol{p}}) < k_2(\hat{\boldsymbol{p}}) + 1$ and $l(\hat{\boldsymbol{p}}) < u(\hat{\boldsymbol{p}})$, we have $k_1(\hat{\boldsymbol{p}}) \leq k_2(\hat{\boldsymbol{p}})$. Define $a = \sum_{j=1}^{k_1(\hat{\boldsymbol{p}})}\hat{p}_j$ and

$$\hat{\boldsymbol{p}}_1 = (a, 0, \cdots, 0, \hat{p}_{k_1(\hat{\boldsymbol{p}})+1}, \hat{p}_{k_1(\hat{\boldsymbol{p}})+2}, \cdots, \hat{p}_m).$$

For short, we denote $R(f, \hat{\boldsymbol{p}}) = \max_{\boldsymbol{p}, \boldsymbol{g}:\hat{\boldsymbol{p}} \propto \boldsymbol{p} \circ \boldsymbol{g}} (f(\hat{\boldsymbol{p}}) - \mathbb{E}_{x \sim \boldsymbol{p}}[x])^2$, which is the maximum regret of $f$ given the empirical distribution $\hat{\boldsymbol{p}}$. We will show $R(f^{PAA}, \hat{\boldsymbol{p}}) \leq R(f^{PAA}, \hat{\boldsymbol{p}}_1)$.

Note for $\hat{\boldsymbol{p}}_1$, we have

$$l(\hat{\boldsymbol{p}}_1) \leq \frac{\frac{1}{q}\sum_{j=1}^{k_1(\hat{\boldsymbol{p}})}\hat{p}_j + \sum_{j=k_1(\hat{\boldsymbol{p}})+1}^{m}j\hat{p}_j}{\frac{1}{q}\sum_{j=1}^{k_1(\hat{\boldsymbol{p}})}\hat{p}_j + \sum_{j=k_1(\hat{\boldsymbol{p}})+1}^{m}\hat{p}_j},$$

and

$$u(\hat{\boldsymbol{p}}_1) \geq \frac{\sum_{j=1}^{k_2(\hat{\boldsymbol{p}})}\hat{p}_j + \frac{1}{q}\sum_{j=k_2(\hat{\boldsymbol{p}})+1}^{m}j\hat{p}_j}{\sum_{j=1}^{k_2(\hat{\boldsymbol{p}})}\hat{p}_j + \frac{1}{q}\sum_{j=k_2(\hat{\boldsymbol{p}})+1}^{m}\hat{p}_j}.$$

So

$$(u(\hat{\boldsymbol{p}}_1) - l(\hat{\boldsymbol{p}}_1)) - (u(\hat{\boldsymbol{p}}) - l(\hat{\boldsymbol{p}}))$$
$$= (l(\hat{\boldsymbol{p}}) - l(\hat{\boldsymbol{p}}_1)) - (u(\hat{\boldsymbol{p}}) - u(\hat{\boldsymbol{p}}_1))$$
$$\geq \left( \frac{\frac{1}{q}\sum_{j=1}^{k_1(\hat{\boldsymbol{p}})} j\hat{p}_j + \sum_{j=k_1(\hat{\boldsymbol{p}})+1}^{m} j\hat{p}_j}{\frac{1}{q}\sum_{j=1}^{k_1(\hat{\boldsymbol{p}})} \hat{p}_j + \sum_{j=k_1(\hat{\boldsymbol{p}})+1}^{m} \hat{p}_j} - \frac{\frac{1}{q}\sum_{j=1}^{k_1(\hat{\boldsymbol{p}})} \hat{p}_j + \sum_{j=k_1(\hat{\boldsymbol{p}})+1}^{m} j\hat{p}_j}{\frac{1}{q}\sum_{j=1}^{k_1(\hat{\boldsymbol{p}})} \hat{p}_j + \sum_{j=k_1(\hat{\boldsymbol{p}})+1}^{m} \hat{p}_j} \right)$$
$$- \left( \frac{\sum_{j=1}^{k_2(\hat{\boldsymbol{p}})} j\hat{p}_j + \frac{1}{q}\sum_{j=k_2(\hat{\boldsymbol{p}})+1}^{m} j\hat{p}_j}{\sum_{j=1}^{k_2(\hat{\boldsymbol{p}})} \hat{p}_j + \frac{1}{q}\sum_{j=k_2(\hat{\boldsymbol{p}})+1}^{m} \hat{p}_j} - \frac{\sum_{j=1}^{k_2(\hat{\boldsymbol{p}})} \hat{p}_j + \frac{1}{q}\sum_{j=k_2(\hat{\boldsymbol{p}})+1}^{m} j\hat{p}_j}{\sum_{j=1}^{k_2(\hat{\boldsymbol{p}})} \hat{p}_j + \frac{1}{q}\sum_{j=k_2(\hat{\boldsymbol{p}})+1}^{m} \hat{p}_j} \right)$$
$$= \frac{\frac{1}{q}\sum_{j=1}^{k_1(\hat{\boldsymbol{p}})} (j-1)\hat{p}_j}{\frac{1}{q}\sum_{j=1}^{k_1(\hat{\boldsymbol{p}})} \hat{p}_j + \sum_{j=k_1(\hat{\boldsymbol{p}})+1}^{m} \hat{p}_j} - \frac{\sum_{j=1}^{k_2(\hat{\boldsymbol{p}})} (j-1)\hat{p}_j}{\sum_{j=1}^{k_2(\hat{\boldsymbol{p}})} \hat{p}_j + \frac{1}{q}\sum_{j=k_2(\hat{\boldsymbol{p}})+1}^{m} \hat{p}_j}$$
$$= \frac{\sum_{j=1}^{k_1(\hat{\boldsymbol{p}})} (j-1)\hat{p}_j}{\sum_{j=1}^{k_1(\hat{\boldsymbol{p}})} \hat{p}_j + q\sum_{j=k_1(\hat{\boldsymbol{p}})+1}^{m} \hat{p}_j} - \frac{\sum_{j=1}^{k_2(\hat{\boldsymbol{p}})} (j-1)\hat{p}_j}{\sum_{j=1}^{k_2(\hat{\boldsymbol{p}})} \hat{p}_j + \frac{1}{q}\sum_{j=k_2(\hat{\boldsymbol{p}})+1}^{m} \hat{p}_j}$$
$$\geq 0.$$

which implies $R(f^{PAA}, \hat{\boldsymbol{p}}) \leq R(f^{PAA}, \hat{\boldsymbol{p}}_1)$.

We also use $\boldsymbol{p}(1)$ to denote the first element of $\boldsymbol{p}$. Define $b = \sum_{j=k_2(\hat{\boldsymbol{p}})+1}^{m} \hat{\boldsymbol{p}}_1(j)$ and $\hat{\boldsymbol{p}}_2 = (\hat{\boldsymbol{p}}_1(1), \hat{\boldsymbol{p}}_1(2), \cdots, \hat{\boldsymbol{p}}_1(k_2(\hat{\boldsymbol{p}}_1)), 0, \cdots, 0, b)$.

Similarly, we can prove $R(f^{PAA}, \hat{\boldsymbol{p}}_1) \leq R(f^{PAA}, \hat{\boldsymbol{p}}_2)$.

We call this one iteration. Note the non-zero entries will not increase during the iteration. We iterate until the non-zero entries do not decrease. We use $\hat{\boldsymbol{p}}'$ to denote the stable distribution after iterations. $\hat{\boldsymbol{p}}'$ has the format of $(\hat{p}_1', 0, \cdots, 0, \hat{p}_l', \hat{p}_{l+1}', \cdots, \hat{p}_r', 0, \cdots, 0, \hat{p}_m')$, where $k_1(\hat{\boldsymbol{p}}') < l$ and $k_2(\hat{\boldsymbol{p}}') > r$.

Define $\hat{\boldsymbol{p}}_3 = (\hat{p}_1', 0, \cdots, 0, 1 - \hat{p}_1' - \hat{p}_m', 0, \cdots, 0, \hat{p}_m')$ where the $l$-th entry is $1 - \hat{p}_1' - \hat{p}_m'$. Define $\hat{\boldsymbol{p}}_4 = (\hat{p}_1', 0, \cdots, 0, 1 - \hat{p}_1' - \hat{p}_m', 0, \cdots, 0, \hat{p}_m')$ where the $r$-th entry is $1 - \hat{p}_1' - \hat{p}_m'$.

$$l(\hat{\boldsymbol{p}}_3) \leq \frac{\frac{1}{q}\sum_{j=1}^{l-1} j\hat{\boldsymbol{p}}_3(j) + \sum_{j=l}^{m} j\hat{\boldsymbol{p}}_3(j)}{\frac{1}{q}\sum_{j=1}^{l-1} \hat{\boldsymbol{p}}_3(j) + \sum_{j=l}^{m} \hat{\boldsymbol{p}}_3(j)} = \frac{\frac{1}{q}\hat{p}_1' + l(1 - \hat{p}_1' - \hat{p}_m') + m\hat{p}_m'}{\frac{1}{q}\hat{p}_1' + (1 - \hat{p}_1')}$$

$$u(\hat{\boldsymbol{p}}_3) \geq \frac{\sum_{j=1}^{r} j\hat{\boldsymbol{p}}_3(j) + \frac{1}{q}\sum_{j=r+1}^{m} j\hat{\boldsymbol{p}}_3(j)}{\sum_{j=1}^{r} \hat{\boldsymbol{p}}_3(j) + \frac{1}{q}\sum_{j=r+1}^{m} \hat{\boldsymbol{p}}_3(j)} = \frac{\hat{p}_1' + l(1 - \hat{p}_1' - \hat{p}_m') + \frac{m}{q}\hat{p}_m'}{(1 - \hat{p}_m') + \frac{1}{q}\hat{p}_m'}$$

If $\hat{p}_1' \leq \hat{p}_m'$, then

$$(u(\hat{\boldsymbol{p}}_3) - l(\hat{\boldsymbol{p}}_3)) - (u(\hat{\boldsymbol{p}}') - l(\hat{\boldsymbol{p}})')$$
$$= (l(\hat{\boldsymbol{p}}') - l(\hat{\boldsymbol{p}}_3)) - (u(\hat{\boldsymbol{p}}') - u(\hat{\boldsymbol{p}}_3))$$
$$\geq \left( \frac{\frac{1}{q}\sum_{j=1}^{k_1(\hat{\boldsymbol{p}}')} j\hat{p}_j' + \sum_{j=k_1(\hat{\boldsymbol{p}}')+1}^{m} j\hat{p}_j'}{\frac{1}{q}\sum_{j=1}^{k_1(\hat{\boldsymbol{p}}')} \hat{p}_j' + \sum_{j=k_1(\hat{\boldsymbol{p}}')+1}^{m} \hat{p}_j'} - \frac{\frac{1}{q}\hat{p}_1' + l(1 - \hat{p}_1' - \hat{p}_m') + m\hat{p}_m'}{\frac{1}{q}\hat{p}_1' + (1 - \hat{p}_1')} \right)$$
$$- \left( \frac{\sum_{j=1}^{k_2(\hat{\boldsymbol{p}}')} j\hat{p}_j' + \frac{1}{q}\sum_{j=k_2(\hat{\boldsymbol{p}}')+1}^{m} j\hat{p}_j'}{\sum_{j=1}^{k_2(\hat{\boldsymbol{p}}')} \hat{p}_j' + \frac{1}{q}\sum_{j=k_2(\hat{\boldsymbol{p}}')+1}^{m} \hat{p}_j'} - \frac{\hat{p}_1' + l(1 - \hat{p}_1' - \hat{p}_m') + \frac{m}{q}\hat{p}_m'}{(1 - \hat{p}_m') + \frac{1}{q}\hat{p}_m'} \right)$$
$$= \left( \frac{\frac{1}{q}\hat{p}_1' + \sum_{j=l}^{r} j\hat{p}_j' + m\hat{p}_m'}{\frac{1}{q}\hat{p}_1' + (1 - \hat{p}_1')} - \frac{\frac{1}{q}\hat{p}_1' + l(1 - \hat{p}_1' - \hat{p}_m') + m\hat{p}_m'}{\frac{1}{q}\hat{p}_1' + (1 - \hat{p}_1')} \right)$$
$$- \left( \frac{\hat{p}_1' + \sum_{j=l}^{r} j\hat{p}_j' + \frac{m}{q}\hat{p}_m'}{(1 - \hat{p}_m') + \frac{1}{q}\hat{p}_m'} - \frac{\hat{p}_1' + l(1 - \hat{p}_1' - \hat{p}_m') + \frac{m}{q}\hat{p}_m'}{(1 - \hat{p}_m') + \frac{1}{q}\hat{p}_m'} \right)$$
$$= \left( \sum_{j=l}^{r} j\hat{p}_j' - l(1 - \hat{p}_1' - \hat{p}_m') \right) \left( \frac{1}{\frac{1}{q}\hat{p}_1' + (1 - \hat{p}_1')} - \frac{1}{(1 - \hat{p}_m') + \frac{1}{q}\hat{p}_m'} \right)$$
$$= \sum_{j=l}^{r} (j-l)\hat{p}_j' \left( \frac{1}{\frac{1}{q}\hat{p}_1' + (1 - \hat{p}_1')} - \frac{1}{(1 - \hat{p}_m') + \frac{1}{q}\hat{p}_m'} \right)$$
$$\geq 0.$$

Similarly, if $\hat{p}_1' > \hat{p}_m'$, we have

$$(u(\hat{\boldsymbol{p}}_4) - l(\hat{\boldsymbol{p}}_4)) - (u(\hat{\boldsymbol{p}}') - l(\hat{\boldsymbol{p}})') \geq 0.$$

So $(u(\hat{\boldsymbol{p}}') - l(\hat{\boldsymbol{p}})') \leq \max\{(u(\hat{\boldsymbol{p}}_3) - l(\hat{\boldsymbol{p}}_3)), (u(\hat{\boldsymbol{p}}_4) - l(\hat{\boldsymbol{p}}_4))\}$, which implies $R(f^{PAA}, \hat{\boldsymbol{p}}') \leq \max\{R(f^{PAA}, \hat{\boldsymbol{p}}_3), R(f^{PAA}, \hat{\boldsymbol{p}}_4)\}$. So $\hat{\boldsymbol{p}}^*$ has at most three non-zero entries.

**Second Step:** Now we prove $\hat{\boldsymbol{p}}^*$ has exactly two non-zero entries. Without loss of generality, we now suppose there are only three ratings: $1, k, m$, where $1 < k < m$. For $\hat{\boldsymbol{p}} = (a, 1 - a - b, b)$, suppose the optimal index for minimum is $k_1(\hat{\boldsymbol{p}}) \in \{1, 2\}$ and the optimal index for maximum is $k_2(\hat{\boldsymbol{p}}) \in \{1, 2\}$. Still we have $k_1(\hat{\boldsymbol{p}}) \leq k_2(\hat{\boldsymbol{p}})$. Define $\hat{\boldsymbol{p}}_5 = (a, 0, 1 - a), \hat{\boldsymbol{p}}_6 = (1 - b, 0, b)$. There are three cases.

**Case 1:** We have $k_1(\hat{\boldsymbol{p}}) = k_2(\hat{\boldsymbol{p}}) = 1$.

$$l(\hat{\boldsymbol{p}}) = \frac{a}{a + q(1-a)} + \frac{q(1 - a - b)}{a + q(1-a)} \times k + \frac{qb}{a + q(1-a)} \times m$$

$$u(\hat{\boldsymbol{p}}) = \frac{qa}{qa + 1 - a} + \frac{1 - a - b}{qa + (1-a)} \times k + \frac{b}{qa + (1-a)} \times m$$

$$l(\hat{\boldsymbol{p}}_5) = \frac{a}{a + q(1-a)} + \frac{q(1-a)}{a + q(1-a)} \times m$$

$$u(\hat{\boldsymbol{p}}_5) = \frac{qa}{qa + (1-a)} + \frac{1-a}{qa + (1-a)} \times m$$

Then

$$(u(\hat{\boldsymbol{p}}_5) - l(\hat{\boldsymbol{p}}_5)) - (u(\hat{\boldsymbol{p}}) - l(\hat{\boldsymbol{p}}))$$
$$= (u(\hat{\boldsymbol{p}}_5) - u(\hat{\boldsymbol{p}})) - (l(\hat{\boldsymbol{p}}_5) - l(\hat{\boldsymbol{p}}))$$
$$= \frac{1 - a - b}{qa + (1-a)} \times k - \frac{q(1 - a - b)}{a + q(1-a)} \times k$$
$$= k(1 - a - b)\left( \frac{1}{qa + (1-a)} - \frac{1}{\frac{a}{q} + (1-a)} \right)$$
$$\geq 0.$$

So $R(f^{PAA}, \hat{p}) \leq R(f^{PAA}, \hat{p}^5)$.

**Case 2:** $k_1(\hat{p}) = k_2(\hat{p}) = 2$. Similarly, we can prove $R(f^{PAA}, \hat{p}) \leq R(f^{PAA}, \hat{p}^6)$.

**Case 3:** $k_1(\hat{p}) = 1, k_2(\hat{p}) = 2$.

$$l(\hat{p}) = \frac{a}{a + q(1-a)} + \frac{q(1-a-b)}{a + q(1-a)} \times k + \frac{qb}{a + q(1-a)} \times m$$

$$u(\hat{p}) = \frac{qa}{b + q(1-b)} + \frac{q(1-a-b)}{b + q(1-b)} \times k + \frac{b}{b + q(1-b))} \times m$$

Define

$$A = \frac{q(1-b)}{b + q(1-b)} + \frac{bm}{b + q(1-b)} - (1 + \frac{qb(m-1)}{a + q(1-a)}),$$

$$B = 1 + \frac{(q(1-a-b)+b)(m-1)}{b + q(1-b)} - (\frac{a}{a + q(1-a)} + \frac{q(1-a)m}{a + q(1-a)}).$$

Since $u(\hat{p}) - l(\hat{p})$ is linear on $k$, we have

$$u(\hat{p}) - l(\hat{p}) \leq \max\{A, B\}.$$

Notice

$$u(\hat{p}_6) = \frac{q(1-b)}{b + q(1-b)} + \frac{bm}{b + q(1-b)}, \quad l(\hat{p}_5) = \frac{a}{a + q(1-a)} + \frac{q(1-a)m}{a + q(1-a)}.$$

So

$$A = u(\hat{p}_6) - (1 + \frac{qb(m-1)}{a + q(1-a)}), \quad B = 1 + \frac{(q(1-a-b)+b)(m-1)}{b + q(1-b)} - l(\hat{p}_5)$$

$$l(\hat{p}_6) - (1 + \frac{qb(m-1)}{a + q(1-a)})$$

$$= \frac{1-b}{qb + 1 - b} + \frac{qbm}{qb + 1 - b} - (1 + \frac{qb(m-1)}{a + q(1-a)})$$

$$= 1 + \frac{qb(m-1)}{qb + 1 - b} - ((1 + \frac{qb(m-1)}{a + q(1-a)}))$$

$$= qb(m-1)(\frac{1}{qb + 1 - b} - \frac{1}{a + q(1-a)})$$

$$= \frac{qb(m-1)}{(qb + 1 - b)(a + q(1-a))}(a + q(1-a) - (qb + 1 - b))$$

$$= \frac{qb(m-1)}{(qb + 1 - b)(a + q(1-a))}(q - 1)(1 - a - b)$$

$$\leq 0$$

$$u(\hat{p}_5) - (1 + \frac{(q(1-a-b)+b)(m-1)}{b + q(1-b)})$$

$$= \frac{qa}{qa + (1-a)} + \frac{(1-a)m}{qa + (1-a)} - \left(1 + \frac{(q(1-a-b)+b)(m-1)}{b + q(1-b)}\right)$$

$$= 1 + \frac{(1-a)(m-1)}{qa + (1-a)} - \left(1 + \frac{(q(1-a-b)+b)(m-1)}{b + q(1-b)}\right)$$

$$= (m-1)(\frac{1-a}{qa + (1-a)} - \frac{q(1-a-b)+b}{b + q(1-b)})$$

$$= (m-1)\left(1 - \frac{qa}{qa + (1-a)} - (1 - \frac{qa}{b + q(1-b)})\right)$$

$$= qa(m-1)(\frac{1}{b + q(1-b)} - \frac{1}{qa + (1-a)})$$

$$= \frac{qa(m-1)}{(qa + (1-a))(b + q(1-b))}(qa + (1-a) - b + q(1-b))$$

$$= \frac{qa(m-1)}{(qa + (1-a))(b + q(1-b))}(1-q)(1-a-b)$$

$$\geq 0.$$

So $A \leq u(\hat{p}_6) - l(\hat{p}_6)$, $B \leq u(\hat{p}_5) - l(\hat{p}_5)$, which implies

$$u(\hat{p}) - l(\hat{p}) \leq \max\{u(\hat{p}_6) - l(\hat{p}_6), u(\hat{p}_5) - l(\hat{p}_5)\},$$

i.e.,

$$R(f^{PAA}, \hat{p}) \leq \max\{R(f^{PAA}, \hat{p}_5), R(f^{PAA}, \hat{p}_6)\}.$$

Putting the three pieces together, we have

$$R(f^{PAA}, \hat{p}) \leq \max\{R(f^{PAA}, \hat{p}_5), R(f^{PAA}, \hat{p}_6)\},$$

which implies $\hat{p}^*$ has exactly two non-zero entries.

**Third Step:** Next, we prove $\hat{p}^* = (\frac{1}{2}, 0, \cdots, 0, \frac{1}{2})$. For $\hat{p} = (a, 0, \cdots, 0, 1-a), 0 \leq a \leq 1, l(\hat{p}) = \frac{a}{a+q(1-a)} + \frac{q(1-a)}{a+q(1-a)} \times m, u(\hat{p}) = \frac{qa}{qa+(1-a)} + \frac{1-a}{qa+(1-a)} \times m$, so

$$u(\hat{p}) - l(\hat{p}) = (m-1) \times \left(\frac{1-a}{qa + (1-a)} - \frac{q(1-a)}{a + q(1-a)}\right)$$

$$= (m-1) \times \frac{(1-q^2)(a-a^2)}{(1-q)^2(a-a^2) + q}$$

$$= (m-1) \times \frac{(1+q)(a-a^2)}{(1-q)(a-a^2) + \frac{q}{1-q}}.$$

Let $x = a - a^2 \in [0, \frac{1}{4}]$, $F(x) = \frac{u(\hat{p})-l(\hat{p})}{m-1} = \frac{(1+q)x}{(1-q)x + \frac{q}{1-q}}$. Then

$$F'(x) = \frac{(1+q)\left((1-q)x + \frac{q}{1-q}\right) - (1-q)(1+q)x}{\left((1-q)x + \frac{q}{1-q}\right)^2}$$

$$= \frac{q(1+q)}{(1-q)\left((1-q)x + \frac{q}{1-q}\right)^2}$$

$$> 0.$$

So $x = \frac{1}{4}$ uniquely maximizes the regret. Since $x = \frac{1}{4} \iff a = \frac{1}{2}$, we prove $\hat{p}^* = (\frac{1}{2}, 0, \cdots, 0, \frac{1}{2})$.

**Fourth Step:** Finally, it is easy to figure out the optimal index for minimum $k_1(\hat{p}^*)$ and the optimal index for maximum $k_2(\hat{p}^*)$ can both be any value in $[m-1]$. Without loss of generality, we set $k_1(\hat{p}^*) = 1, k_2(\hat{p}^*) = m - 1$. So the corresponding worst pair of information structure is

- $\theta_1 = (\boldsymbol{p}_1 = [\frac{1}{q+1}, 0, \cdots, 0, \frac{q}{q+1}], \boldsymbol{g}_1 = [q, 1, \cdots, 1, 1])$
- $\theta_2 = (\boldsymbol{p}_2 = [\frac{q}{q+1}, 0, \cdots, 0, \frac{1}{q+1}], \boldsymbol{g}_2 = [1, 1, \cdots, 1, q])$

$\square$

THEOREM A.4. *When* $m = o\left(\left(\frac{n}{\ln n}\right)^{\frac{1}{4}}\right)$, *PAA is* $O\left(m^2\sqrt{\frac{\ln n}{n}}\right)$-*optimal. That is, for any aggregator* $f$,

$$\max_{\boldsymbol{p},\boldsymbol{g}} R(f^{PAA}, \boldsymbol{p}, \boldsymbol{g}) \le \max_{\boldsymbol{p},\boldsymbol{g}} R(f, \boldsymbol{p}, \boldsymbol{g}) + O\left(m^2\sqrt{\frac{\ln n}{n}}\right).$$

PROOF. For clarity, we define $R_n(f^{PAA}, \boldsymbol{p}, \boldsymbol{g})$, which is the regret of PAA when the sample size is $n$. Suppose $f_n^*$ is the optimal aggregator when sample size is $n$. It is equivalent to prove

$$\max_{\boldsymbol{p},\boldsymbol{g}} R_n(f^{PAA}, \boldsymbol{p}, \boldsymbol{g}) \le \max_{\boldsymbol{p},\boldsymbol{g}} R_n(f_n^*, \boldsymbol{p}, \boldsymbol{g}) + O\left(m^2\sqrt{\frac{\ln n}{n}}\right)$$

Use $R_n(f^*)$ to denote the optimal regret when the sample size is $n$, and $R^*$ is the optimal regret when $n \to \infty$. First Notice $R_n(f^*)$ is a non-increasing function with respect to $n$ since we can always choose the aggregator which only uses part of the ratings, so $R_n(f^*) \ge R^*$. Then We bound the difference between $R_n(f^{PAA}, \boldsymbol{p}, \boldsymbol{g})$ and $R^*$ by concentration inequality.

For any $(\boldsymbol{p}, \boldsymbol{g})$, when $n \to \infty$, we obtain $\hat{\boldsymbol{p}} = \frac{\boldsymbol{p}\boldsymbol{g}}{\sum_j p_j g_j}$ and

$$R_\infty(f^{PAA}, \boldsymbol{p}, \boldsymbol{g}) = (f^{PAA}(\hat{\boldsymbol{p}}) - \mathbb{E}_{x \sim \boldsymbol{p}}[x])^2 = \left(\frac{u(\hat{\boldsymbol{p}}) + l(\hat{\boldsymbol{p}})}{2} - \mathbb{E}_{x \sim \boldsymbol{p}}[x]\right)^2$$

In the finite case, we get a noisy distribution $\hat{\boldsymbol{p}}'$. We use $n_r$ to denote the number of people whose rating is $r$ and $\hat{n}_r$ to denote the number of people who actually report $r$.

$$R_n(f^{PAA}, \boldsymbol{p}, \boldsymbol{g}) = \mathbb{E}[(f^{PAA}(\hat{\boldsymbol{p}}') - \frac{\sum_r r n_r}{n}] = \mathbb{E}[(\frac{u(\hat{\boldsymbol{p}}') + l(\hat{\boldsymbol{p}}')}{2} - \frac{\sum_r r n_r}{n})^2]$$

By chernoff bound, we have

$$\Pr[|n_r - E[n_r]| \ge \sqrt{n \ln n}] \le 2e^{-\frac{2n\ln n}{n}} = \frac{2}{n^2}$$

$$\Pr[|\hat{n}_r - E[\hat{n}_r]| \ge \sqrt{n \ln n}] \le 2e^{-\frac{2n\ln n}{n}} = \frac{2}{n^2}$$

$$\Pr[|\sum_r \hat{n}_r - E[\sum_r \hat{n}_r]| \ge \sqrt{n \ln n}] \le 2e^{-\frac{2n\ln n}{n}} = \frac{2}{n^2}$$

Define event $A_r = \{|n_r - E[n_r]| \ge \sqrt{n \ln n}\}$, event $A = \bigvee_{r=1}^m A_r$, event $B_r = \{|\hat{n}_r - E[\hat{n}_r]| \ge \sqrt{n \ln n}\}$, event $B = \bigvee_{r=1}^m B_r$, event $C = \{|\sum_r \hat{n}_r - E[\sum_r \hat{n}_r]| \ge \sqrt{n \ln n}\}$. By union bound, we have

$$\Pr[A \vee B \vee C] \le \frac{4m+2}{n^2} \le \frac{1}{n}$$

**Case 1:** $A$ happens or $B$ happens or $C$ happens.

$$\mathbb{E}\left[\left(\frac{u(\hat{\boldsymbol{p}}') + l(\hat{\boldsymbol{p}}')}{2} - \frac{\sum_r r n_r}{n}\right)^2 \mid A \vee B \vee C\right] = O(m^2)$$

**Case 2:** None of $A, B, C$ happens. We have

$$\left|\hat{p}_r' - \hat{p}_r\right| = \left|\frac{\hat{n}_r}{\sum_j \hat{n}_j} - \frac{p_r g_r}{\sum_j p_j g_j}\right|$$

$$= \left|\frac{E[\hat{n}_r] + O(\sqrt{n \ln n})}{E[\sum_j \hat{n}_j] + O(\sqrt{n \ln n})} - \frac{p_r g_r}{\sum_j p_j g_j}\right|$$

$$= \left|\frac{p_r g_r n + O(\sqrt{n \ln n})}{\sum_j p_j g_j n + O(\sqrt{n \ln n})} - \frac{p_r g_r}{\sum_j p_j g_j}\right|$$

$$= O\left(\sqrt{\frac{\ln n}{n}}\right)$$

When $n$ is large enough, this error will not influence the optimal index. So

$$\left|l(\hat{\boldsymbol{p}}') - l(\hat{\boldsymbol{p}})\right|$$

$$= \left|\frac{\frac{1}{q}\sum_{j=1}^{k_1} j\hat{p}_j' + \sum_{j=k_1+1}^m j\hat{p}_j'}{\frac{1}{q}\sum_{j=1}^{k_1} \hat{p}_j' + \sum_{j=k_1+1}^m \hat{p}_j'} - \frac{\frac{1}{q}\sum_{j=1}^{k_1} j\hat{p}_j + \sum_{j=k_1+1}^m j\hat{p}_j}{\frac{1}{q}\sum_{j=1}^{k_1} \hat{p}_j + \sum_{j=k_1+1}^m \hat{p}_j}\right|$$

$$= \left|\frac{\frac{1}{q}\sum_{j=1}^{k_1} j\hat{p}_j + \sum_{j=k_1+1}^m j\hat{p}_j + O\left(m^2\sqrt{\frac{\ln n}{n}}\right)}{\frac{1}{q}\sum_{j=1}^{k_1} \hat{p}_j + \sum_{j=k_1+1}^m \hat{p}_j + O\left(m\sqrt{\frac{\ln n}{n}}\right)} - \frac{\frac{1}{q}\sum_{j=1}^{k_1} j\hat{p}_j + \sum_{j=k_1+1}^m j\hat{p}_j}{\frac{1}{q}\sum_{j=1}^{k_1} \hat{p}_j + \sum_{j=k_1+1}^m \hat{p}_j}\right|$$

$$= O\left(m^2\sqrt{\frac{\ln n}{n}}\right)$$

Similarly, $\left|u(\hat{\boldsymbol{p}}') - u(\hat{\boldsymbol{p}})\right| = O\left(m^2\sqrt{\frac{\ln n}{n}}\right)$, so

$$\left|\frac{u(\hat{\boldsymbol{p}}') + l(\hat{\boldsymbol{p}}')}{2} - \frac{u(\hat{\boldsymbol{p}}) + l(\hat{\boldsymbol{p}})}{2}\right| = O\left(m^2\sqrt{\frac{\ln n}{n}}\right)$$

Notice

$$\left|\frac{\sum_r r n_r}{n} - \mathbb{E}_{x \sim \boldsymbol{p}}[x]\right| = \left|\frac{\sum_r r n_r}{n} - \frac{\sum_r r E[n_r]}{n}\right| = \left|\frac{\sum_r r(n_r - E[n_r])}{n}\right| = O\left(m^2\sqrt{\frac{\ln n}{n}}\right)$$

Then we have

$$\mathbb{E}\left[\left(\frac{u(\hat{\boldsymbol{p}}') + l(\hat{\boldsymbol{p}}')}{2} - \sum_r r n_r\right)^2 \mid \neg(A \vee B \vee C)\right]$$

$$= \left(\frac{u(\hat{\boldsymbol{p}}) + l(\hat{\boldsymbol{p}})}{2} - \mathbb{E}_{x \sim \boldsymbol{p}}[x] + O\left(m^2\sqrt{\frac{\ln n}{n}}\right)\right)^2$$

$$= R_\infty(f^{PAA}, \boldsymbol{p}, \boldsymbol{g}) + O\left(m^2\sqrt{\frac{\ln n}{n}}\right)$$

Putting the two pieces together, we have

$$R_n(f^{PAA}, \boldsymbol{p}, \boldsymbol{g})$$

$$= \Pr[A \vee B \vee C] \, \mathbb{E}\left[ (\frac{u(\hat{\boldsymbol{p}}') + l(\hat{\boldsymbol{p}}')}{2} - \frac{\sum_r r n_r}{n})^2 \mid A \vee B \vee C \right]$$

$$+ \Pr[\neg(A \vee B \vee C)] \, \mathbb{E}\left[ (\frac{u(\hat{\boldsymbol{p}}') + l(\hat{\boldsymbol{p}}')}{2} - \frac{\sum_r r n_r}{n})^2 \mid \neg(A \vee B \vee C) \right]$$

$$= \Pr[A \vee B \vee C] \, O(m^2) + \Pr[\neg(A \vee B \vee C)] \left( R_n(f^{PAA}, \boldsymbol{p}, \boldsymbol{g}) + O\left(m^2 \sqrt{\frac{\ln n}{n}}\right) \right)$$

$$\leq \frac{m^2}{n} + (1 - \frac{1}{n}) \left( R_n(f^{PAA}, \boldsymbol{p}, \boldsymbol{g}) + O\left(m^2 \sqrt{\frac{\ln n}{n}}\right) \right)$$

$$= R_\infty(f^{PAA}, \boldsymbol{p}, \boldsymbol{g}) + O\left(m^2 \sqrt{\frac{\ln n}{n}}\right)$$

Since for any $\boldsymbol{p}, \boldsymbol{g}$ the inequality holds, we have

$$\max_{\boldsymbol{p}, \boldsymbol{g}} R_n(f^{PAA}, \boldsymbol{p}, \boldsymbol{g}) \leq \max_{\boldsymbol{p}, \boldsymbol{g}} R_\infty(f^{PAA}, \boldsymbol{p}, \boldsymbol{g}) + O\left(m^2 \sqrt{\frac{\ln n}{n}}\right)$$

So for any sample size $n$,

$$\max_{\boldsymbol{p}, \boldsymbol{g}} R_n(f^{PAA}, \boldsymbol{p}, \boldsymbol{g}) \leq \max_{\boldsymbol{p}, \boldsymbol{g}} R_\infty(f^{PAA}, \boldsymbol{p}, \boldsymbol{g}) + O\left(m^2 \sqrt{\frac{\ln n}{n}}\right)$$

$$= R^* + O\left(m^2 \sqrt{\frac{\ln n}{n}}\right)$$

$$\leq R_n(f^*) + O\left(m^2 \sqrt{\frac{\ln n}{n}}\right)$$

$$= \max_{\boldsymbol{p}, \boldsymbol{g}} R(f_n^*, \boldsymbol{p}, \boldsymbol{g}) + O\left(m^2 \sqrt{\frac{\ln n}{n}}\right)$$

The first equality holds since $f^{PAA}$ is the optimal aggregator when $n \to \infty$.

□

## B  Omitted Figures

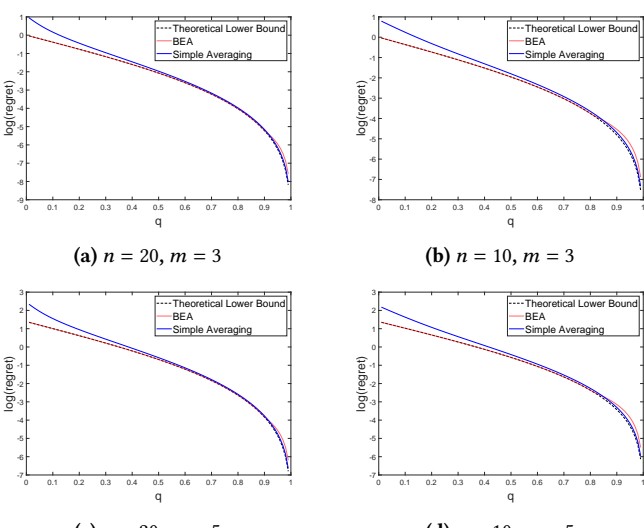

(a) $n = 20, m = 3$

(b) $n = 10, m = 3$

(c) $n = 20, m = 5$

(d) $n = 10, m = 5$

**Figure 11: Simple averaging vs. BEA for different sample size $n$ and the number of rating categories $m$. The x-axis is the lower bound of the participation probability, $q$, and the y-axis is the natural logarithm of the regret. The regret of BEA almost matches the theoretical lower bound for a wide range of $q$.**

Figure 11 are the regret of different aggregators with $q$ ranging from 0.01 to 0.99 with step 0.01. Notice when $q$ is large (the threshold is around 0.82 for $n = 10$ and 0.91 for $n = 20$), the regret of BEA deviates from the theory lower bound and even exceeds the regret of simple average aggregator when $q$ is larger. This is because the worst information structure for BEA changes from the pair in Lemma 3.2 to

$$\theta = (\boldsymbol{p} = [b, 0, \cdots, 0, 1 - b], \boldsymbol{g}_1 = [q, q, \cdots, q, q])$$

where $b$ is dependent with $n$ and $q$.

Received 20 February 2007; revised 12 March 2009; accepted 5 June 2009

