# OpenReview forum: "Mitigating the Participation Bias by Balancing Extreme Ratings"
_ACM.org/TheWebConf/2025/Conference — WWW 2025 Oral_

### Official Review · Reviewer_m1tU · 2024-11-23

**Novelty:** 5
**Technical Quality:** 5

**Review:**

The authors address the problem of robust rating aggregation under participation bias (where ratings do not represent the views of all customers). They demonstrate that relying solely on empirical averages from received reviews is biased and propose an approach to computing average ratings. To evaluate aggregation methods in the presence of participation bias, they explore two optimal aggregator settings: Balanced Extremes Aggregator (BEA) for known numbers of raters and Probability Averaging Algorithm (PAA) for unknown numbers.

BEA aggregates two score: one for observed ratings and another for unobserved ratings. The latter depends on a parameter that is determined by solving the maximization problem formulated in Lemma 3.2, which involves observed data (ratings). However, it is unclear how to solve this problem analytically or numerically, particularly with regards to convergence rates and online solvability.  While not a primary focus, providing more details on the solution algorithm for problem in Corollary 3.3  and its performance characteristics (e.g., convergence rates, online solvability) would enhance the paper. The authors' evaluation shows that BEA's performance closely approaches the theoretical lower bound, with its outperforming simple averaging.

PAA aggregates two score: one for the lower bound of the true expectation given the empirical distribution of following ratings and another for upper bound. Under asymptotic conditions (n → ∞), PAA is shown to be optimal. They further establish that PAA is applicable for the finite case and achieves near-optimality, with an error bound that scales with both the sample size (n) and number of rating categories (m).

This approach offers an attractive combination of novelty, ease of implementation, and theoretical soundness. While not a primary focus, providing more details on the solution algorithm for problem in Corollary 3.3  and its performance characteristics (e.g., convergence rates, online solvability) would enhance the paper. Despite having access to a vast array of open datasets (there is the simple setting based only on ratings in this study) the authors miss an opportunity to rigorously evaluate BEA's and PAA's efficacy on more complex configurations, sticking instead to a relatively simple setup in their experiments.

**Questions:**

While not a primary focus, providing more details on the solution algorithm for problem in Corollary 3.3  and its performance characteristics (e.g., convergence rates, online solvability) would enhance the paper. Is an online solution possible for a large number of agents?

Despite having access to a vast array of open datasets (there is the simple setting based only on ratings in this study) the authors miss an opportunity to rigorously evaluate BEA's and PAA's efficacy on more complex configurations, sticking instead to a relatively simple setup in their experiments. Are you planning more large-scale experiments?

**Reviewer Confidence:**

4: The reviewer is certain that the evaluation is correct and very familiar with the relevant literature

**Scope:**

4: The work is relevant to the Web and to the track, and is of broad interest to the community

---

### Official Review · Reviewer_EAXw · 2024-11-29

**Novelty:** 5
**Technical Quality:** 6

**Review:**

The authors study the problem of robustly aggregating ratings under participation bias.
Under a simple model of participation bias, they provide aggregators that robustly aggregate an average rating in two settings.
The authors theoretically and numerically evidence the optimality of their aggregators.

Pros:
- Using a robust aggregation approach is a creative and novel way to address participation bias in rating aggregation.
- The paper addresses two important settings, when the sample size is known and when it isn't.
- The paper provides optimal aggregators for these two settings.
- There is potential for important future work.
  The model of participation bias is naive, but it makes a good starting point.
  Future work can explore robust aggregation in more realistic and necessarily more complicated models of participation bias.

Cons:
- The paper considers a simple model of participation bias, the motivation, advantages, and limitations of this model could be developed further.
  (Note, I think this is a good model to begin with, but the rationale should be better developed while acknowledging its limitations)
- As the authors point out, the paper considers a restricted setting (i.e. ratings on a single item), the motivation, advantages, and limitations could be developed further.
  (I think this is a good setting to begin with, but the rationale should be better developed while acknowledging its limitations)
- The CCS concepts don't fit the paper.
  Game theory and mechanism design don't describe the analysis well.
  The agents aren't acting strategically, and the aggregators are not mechanisms in the sense of eliciting actions/information from strategic agents.
- The novel aggregators are only compared to naive averaging (i.e., no adjustment for participation bias).
  It is unclear how well the aggregators perform relative to other non-naive aggregation methods that address participation bias in some way.
  In particular, it would be helpful for practitioners to know what they are trading off for robustness and how when they consider various aggregators in the literature.

Errata:
- Line 306: "casual" --> causal
- Line 347: "casual" --> causal

**Questions:**

- Why this model of participation bias?
- Is there any empirical evidence to support this particular model of participation bias?
- Is there potential to extend or generalize this model to more realistically capture forms of participation bias?
- On line 127: The authors state: "We assume the ratings are independent. Otherwise, the actual ratings may contain systematic biases, making it impossible for any aggregator to recover the true expectation even given all ratings."
  I'm confused by this statement because: a) your model assumes a form of systematic biases; and b) it seems like many kinds of systematic bias can be corrected for with suitable knowledge of the systematic bias.
  Could you clarify what you meant by this statement?
- I think that using a robust aggregation approach also means that there is more room for divergence in the model from reality.
  Is this so and what are your thoughts?
- Did you numerically compare your aggregators against any more sophisticated strategies?
- Line 136: Maybe I missed it, but I don't see where captial bold X is defined.

**Reviewer Confidence:**

3: The reviewer is confident but not certain that the evaluation is correct

**Scope:**

4: The work is relevant to the Web and to the track, and is of broad interest to the community

---

### Official Review · Reviewer_MD95 · 2024-11-29

**Novelty:** 4
**Technical Quality:** 4

**Review:**

This paper addresses the challenge of participation bias in rating systems where raters are more likely to provide feedback when they have extreme opinions. The authors propose two novel aggregation methods to estimate the true average rating: the Balanced Extremes Aggregator (BEA) for scenarios with known sample sizes, and the Polarizing-Averaging Aggregator (PAA) for cases with unknown sample sizes.

# Pros
- Addresses a real and widespread problem in modern rating systems
- Clear problem formulation with mathematical rigor
- Provides two practical solutions for different scenarios

# Cons
- Some assumptions seem too strong, e.g, homogeneous, independence
- Limited discussion of empirical validation on real-world datasets

**Questions:**

See Cons

**Reviewer Confidence:**

1: The reviewer's evaluation is an educated guess

**Scope:**

3: The work is somewhat relevant to the Web and to the track, and is of narrow interest to a sub-community

---

### Official Review · Reviewer_1nS9 · 2024-12-02

**Novelty:** 5
**Technical Quality:** 5

**Review:**

The paper addresses a practical and significant issue: participation bias in rating scenarios, where some raters do not participate in the rating process, leading to potential distortions in the aggregated ratings. To tackle this issue, the authors propose two methods tailored to different sample scenarios:

1. Balanced Extremes Aggregator (BEA) for scenarios with a known sample size,
2. Polarizing-Averaging Aggregator (PAA) for scenarios with an unknown sample size.

The paper is well-structured and easy to follow, with clearly defined methods supported by rigorous theoretical derivations.

Pros
1. The paper investigates an interesting and meaningful problem. Rating aggregation, as described, has practical value across multiple application domains. The proposed methods are designed to handle both known and unknown sample size scenarios, increasing their relevance and applicability.
2.The problem definition, method explanations, and theoretical derivations are presented clearly. The process of addressing the problem, especially through adjusting and combining extreme ratings, is well-articulated.
3.The paper evaluates the robustness of BEA and PAA under varying participation probabilities q, showcasing their ability to handle low participation rates.

Cons
1.The paper lacks validation on real-world datasets, which limits the practical persuasiveness of the proposed methods. Rating scenarios are widely encountered in real-life systems.
2.Both BEA and PAA depend on the hyperparameter q. However, the paper does not discuss how q can be estimated in practice. This could affect the applicability of the methods.
3.In Figure 5, the experiments are conducted only for n=10 and n=20. These small sample sizes are quite limited. Additionally, the paper only compares the proposed methods with simple averaging as the baseline, lacking comparisons with other general aggregation methods.

**Questions:**

1.While the paper evaluates BEA and PAA using simulated data, would it be possible to validate the performance of Balancing Extreme Ratings using real-world datasets?
2.The comparison experiments only involve simple averaging. How would the proposed methods perform when compared with other aggregation methods?
3.In Figure 5, the experiments consider only small sample sizes (n=10 or n=20). For larger sample sizes, such as in the hundreds or thousands, how would the robustness of BEA and PAA change? Would their performance scale effectively with increasing n?

**Reviewer Confidence:**

3: The reviewer is confident but not certain that the evaluation is correct

**Scope:**

4: The work is relevant to the Web and to the track, and is of broad interest to the community

---

### Official Review · Reviewer_vrzJ · 2024-12-02

**Novelty:** 6
**Technical Quality:** 6

**Review:**

Summary

This paper studies the problem of how to aggregate ratings (or more generally, any discrete real-valued signal) under partial censorship with the probability that a given rating is reported depends on the rating itself. Specifically, the authors consider the setting where there are n ratings, drawn from some distribution D with support {1, 2, ..., m}. If the ith rating is x_i, it is censored with some probability g(x_i) (in [q, 1]) and otherwise revealed. The learner is provided with all the uncensored ratings and wants to produce an estimate for the average mu of D, without knowing g or the original distribution. Specifically, they want to minimize their worst-case “regret”; the worst-case (over all choices of g and D) of the gap between their expected square error and that of the omniscient predictor who predicts (1/n) * sum (x_i).

The authors consider two settings. In the first, the total number of ratings n (and thus the number of censored ratings) is known to the learner. In this case, the authors show that a specific aggregator they call the Balanced Extreme Aggregator (BEA) is optimal. BEA simply averages the empirical mean of the uncensored ratings and a specific robust estimate of the censored ratings.

In the second, the total number of original ratings n is unknown. Here they propose an aggregator called Polarizing Average Aggregator (PAA) which first produces two new datasets via subsampling: one where they throw out every element > than some lower bound threshold with probability q, the other where they throw out every element < than some upper bound threshold with probability q. PAA then returns the average of the empirical means of both thresholds. The authors show that in the limit as n -> infinity PAA is the optimal aggregator (and within O~(m^2/sqrt(n)) regret of the optimal aggregator for finite n).


Evaluation

I enjoyed reading this paper. The authors explore a very natural and clean statistical problem, and provide a fairly complete solution to this. In fact, this seems like such a natural statistical problem (inference under a general notion of censorship) that I am a little surprised it hasn’t been studied before. The results were clearly presented and easy to understand.

My only suggestion is that it would have been nice to discuss / explore some theoretical directions a little bit more. For example, how does the worst-case regret of BEA and PAA scale asymptotically with m and n (there is a discussion of the asymptotic gap between PAA and the optimal aggregator in the unknown n case, but it would be interesting to understand how the regret scales with n too; for BEA it can technically be read off of the expression in Lemma 3.2 but it quite unwieldy)? Also, does this extend to D with infinite support (e.g., what if instead of support {1, 2, ..., m}, the support was [0, 1])? multi-dimensional aggregators?

**Questions:**

Feel free to reply to any of the questions in the above review.

**Reviewer Confidence:**

3: The reviewer is confident but not certain that the evaluation is correct

**Scope:**

3: The work is somewhat relevant to the Web and to the track, and is of narrow interest to a sub-community